# Genetic and neuronal mechanisms governing the sex-specific interaction between sleep and sexual behaviors in *Drosophila*

Dandan Chen[1], Divya Sitaraman[2,3,4], Nan Chen[2], Xin Jin[3], Caihong Han[1], Jie Chen[1], Mengshi Sun[1], Bruce S. Baker[2], Michael N. Nitabach[2,3,5,6] & Yufeng Pan[1,7]

Animals execute one particular behavior among many others in a context-dependent manner, yet the mechanisms underlying such behavioral choice remain poorly understood. Here we studied how two fundamental behaviors, sex and sleep, interact at genetic and neuronal levels in *Drosophila*. We show that an increased need for sleep inhibits male sexual behavior by decreasing the activity of the male-specific P1 neurons that coexpress the sex determination genes *fru*M and *dsx*, but does not affect female sexual behavior. Further, we delineate a sex-specific neuronal circuit wherein the P1 neurons encoding increased courtship drive suppressed male sleep by forming mutually excitatory connections with the *fru*M-positive sleep-controlling DN1 neurons. In addition, we find that FRUM regulates male courtship and sleep through distinct neural substrates. These studies reveal the genetic and neuronal basis underlying the sex-specific interaction between sleep and sexual behaviors in *Drosophila*, and provide insights into how competing behaviors are co-regulated.

[1] The Key Laboratory of Developmental Genes and Human Disease, Institute of Life Sciences, Southeast University, Nanjing 210096, China. [2] Janelia Research Campus, Howard Hughes Medical Institute, 19700 Helix Drive, Ashburn, Virginia 20147, USA. [3] Department of Cellular and Molecular Physiology, Yale School of Medicine, New Haven, Connecticut 06520, USA. [4] Department of Psychological Sciences, University of San Diego, San Diego, California 92110, USA. [5] Department of Genetics, Yale School of Medicine, New Haven, Connecticut 06520, USA. [6] Kavli Institute for Neuroscience, Yale School of Medicine, New Haven, Connecticut 06520, USA. [7] Co-innovation Center of Neuroregeneration, Nantong University, Nantong, Jiangsu 226001, China. Dandan Chen and Divya Sitaraman contributed equally to this work.   Correspondence and requests for materials should be addressed to D.S. (email: dsitaraman@sandiego.edu) or to M.N.N. (email: michael.nitabach@yale.edu) or to Y.P. (email: pany@seu.edu.cn)

A fundamental tenant of biology is that organisms sense their environment, and, in response to sensory inputs, alter their physiology and behavior in ways that may be beneficial to the organism[1]. When an organism is faced with more than one stimulus in the context of distinct behavioral states, multiple decision-making processes are involved in making appropriate behavioral choice[1, 2]. Behavioral choice in the context of internal state and external stimuli has been studied in both invertebrates and vertebrates[3, 4], but how competing behaviors interact at the genetic and neuronal levels to ensure appropriate decision-making is still poorly understood.

*Drosophila melanogaster*, like other animals, engages in adaptive innate behaviors such as reproduction and sleep, and the molecular and neuronal mechanisms underlying these behaviors have been intensively studied for decades (see reviews on courtship[5–7] and sleep[8–11]). In males, the courtship behavior is largely controlled by the *fruitless* (*fru*) and *doublesex* (*dsx*) genes. The male-specific proteins (FRU$^M$) derived from the P1 promoter of the *fru* gene (*fru$^M$*) are necessary for innate courtship, and are sufficient for some aspects of courtship[12–14]. The sex-specific products of the *dsx* gene (DSX$^M$ in males and DSX$^F$ in females) are involved in experience-dependent courtship in the absence of FRU$^M$ [15], and courtship intensity and sine song production in the presence of FRU$^M$ [16–18]. FRU$^M$ and DSX$^M$ are expressed in a dispersed subset of *ca.* 2000 and 700 neurons, respectively, which partially overlap[13, 14, 19–22].

In the last two decades *Drosophila* has emerged as a promising model to study the molecular and circuit basis of sleep regulation. Many efforts have been made to identify the neuronal substrates controlling sleep behavior in flies, e.g., the mushroom bodies[23, 24], mushroom body output neurons[25], the Fan-shaped body[26], and the DN1 circadian clock neurons[27]. Although sleep is a sexually dimorphic behavior[28, 29], the sex-specific mechanisms of sleep regulation remain unknown.

Using the amenability of *Drosophila* as a model system for genetic, behavioral, and physiological approaches, we sought to explore the interaction, at various levels, between sexual and sleep behaviors, in order to understand how these competing behaviors are co-regulated to ensure appropriate behavioral choice.

In this study, we show that sleep and sexual behaviors interact in a sex-specific manner. Sleep-deprived male flies display reduced courtship to females, but sleep-deprived female flies are equally receptive to courting males. Furthermore, sexually aroused males have poor sleep, but aroused females sleep more. We further identify the neural substrates involving the male-specific *fru$^M$*-expressing P1 neurons and *dsx*-expressing P1 neurons[30] and *fru$^M$*-expressing DN1 neurons[31, 32] that control such sex-specific interaction between sex and sleep. Specifically, we find that the male-specific P1 courtship command neurons are inhibited by sleep deprivation (SD), and they control sleep in male flies by forming mutually excitatory synaptic connections with sleep-regulating DN1 neurons. Our studies also identify the key genes that control sexually dimorphic sleep behavior in male and female flies paving way for future studies. Together, our results provide a novel framework for investigating genetic and neuronal mechanisms governing the interaction between sleep and sexual behaviors.

## Results

**SD inhibits male courtship.** To determine whether sleep and sexual behavior in flies influence one another, we first asked whether SD alters the display of sexual behaviors. Wild-type Canton-S male and female flies were sleep-deprived by intermittent mechanical perturbation[33], and then assayed for male courtship (Fig. 1a) and female receptivity (Supplementary Fig. 1), respectively. As reported in a recent study[34], we found

that wild-type male flies that were sleep-deprived by mechanical perturbation for 12 h during nighttime (ZT12 to ZT0 of the subsequent day, ZT0 is lights-on, and ZT12 is lights-off in 12 h:12 h light:dark condition) do not show obvious courtship deficits (Fig. 1a). However, we found that courtship by male flies that were sleep-deprived for the same amount of time (12 h), but from ZT16 to ZT4, was significantly reduced (Fig. 1a). Furthermore, SD of males for 16 h, either from ZT8 to ZT0 or from ZT12 to ZT4, severely impaired male courtship (Fig. 1a). Males that were sleep-deprived for 16 h have reduced walking speed indicative of increased sleep need, but they do move around during the 10-min test (Supplementary Fig. 2). These results indicate that both duration and time-of-day when male flies are sleep-deprived play a critical role in sleep-loss-induced courtship deficits.

To investigate whether SD-induced courtship deficit in male flies is a general effect of mechanical perturbation, we deprived males of sleep for 8 h from ZT20 to ZT4, using the same amount of mechanical perturbation with the above 16-h SD (30 s/min shaking for 8 h vs. 15 s/min shaking for 16 h). The 16-h SD severely reduced male sleep and induced sleep rebound after SD (Fig. 1b, c); in contrast, the 8-h SD reduced sleep but did not induced sleep rebound (Fig. 1d, e). Consistent with this, the 8-h SD male flies court much more than the 16-h SD male flies (Fig. 1f). These data suggest that it is the prolonged sleep loss, rather than other possible effects of mechanical perturbation, that impairs male courtship.

Unlike males, pre-mating behaviors in females are less demanding and include slowing down and stopping to allow copulation. In contrast to the above results in males, wild-type females that were sleep-deprived for 12 (ZT16 to ZT4), 16 (ZT12 to ZT4), or 20 (ZT8 to ZT4) hours did not significantly reduce their receptivity to courting males (Supplementary Fig. 1). Taken together, these results indicate that SD suppresses sexual behavior in male flies, but not in female flies.

**Sex-promoting neurons regulate sleep.** To further address the relationship between sleep and sex, we asked whether manipulating neurons that involved in sexual behaviors would affect sleep in either sex. It has been shown that a subset of *fru$^M$*- and *dsx*-expressing neurons, termed P1 neurons, promotes male sexual behavior by integrating chemosensory information[35–40]. More recently, P1 neurons have also been implicated in aggression[41, 42]. Thus, we tested whether activating or inhibiting P1 neurons directly influences sleep. We first used an intersectional strategy (Fig. 2a) to specifically express the temperature-sensitive cation channel dTRPA1 in P1 neurons as previously described (*LexAop2-FlpL/R71G01-LexA;UAS>stop>dTrpA1/dsx$^{GAL4}$*)[37]. Male flies sleep less at 28.5 °C (when P1 neurons are activated) than at 21.5 °C, while control males lacking *R71G01-LexA*(*LexAop2-FlpL/+; UAS>stop>dTrpA1/dsx$^{GAL4}$*) sleep the same amount at 28.5 and 21.5 °C (Fig. 2b, c). The sleep reduction in P1-activated male flies is not due to changes in general locomotor activity, as the experimental and control genotypes are equally active while awake (Supplementary Fig. 3). Detailed analysis revealed that P1-activated male flies sleep less during both daytime and night, and they have increased number of sleep bouts and decreased sleep bout duration, indicative of sleep fragmentation (Supplementary Fig. 3). A second intersectional genetic strategy to target P1 neurons using a *split-GAL4* line (*R15A01-AD; R71G01-DBD*; Fig. 2e)[41] similarly results in decrease in sleep when P1 neurons are conditionally activated (Fig. 2f).

Since activation of P1 neurons decreases sleep and increased arousal in male flies, we wanted to determine whether P1 activity

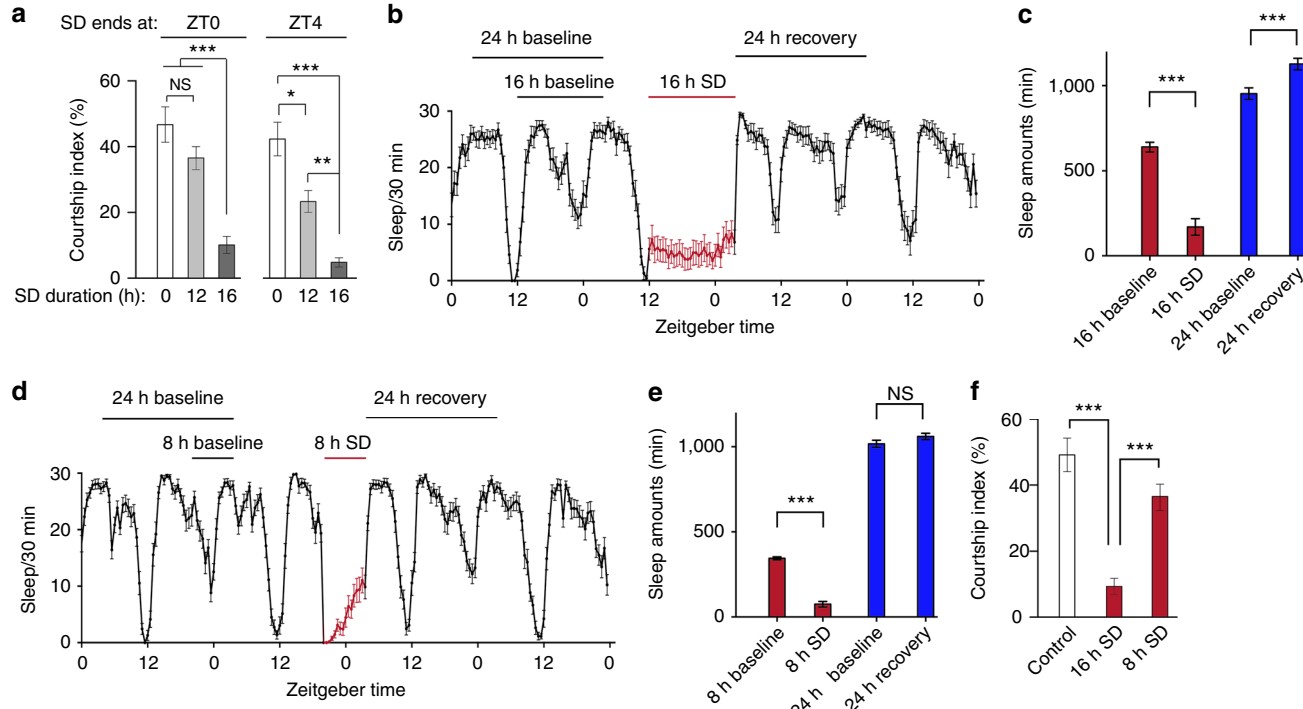

**Fig. 1** SD suppresses male courtship. **a** Courtship indices of males that had been sleep-deprived for 12 h (*light gray*) or 16 h (*dark gray*) during indicated periods. $n = 24$ for each. $*p < 0.05$, $**p < 0.01$, and $***p < 0.001$, one-way ANOVA. **b** Detailed sleep profile of male flies with 16-h SD (15 s/min shaking). **c** The 16-h SD results in 468-min sleep loss during SD and induces post-SD sleep rebound. The start point of baseline was chosen at the same ZT the day before SD. $n = 32$. $***p < 0.001$, unpaired *t*-test. **d** Detailed sleep profile of male flies with 8-h SD (30 s/min shaking), but the amount of mechanical perturbation is the same as the above 16-h SD. **e** The 8-h SD results in 269-min sleep loss during SD, and does not induce post-SD sleep rebound. $n = 32$. N.S., not significant; $***p < 0.001$, unpaired *t*-test. **f** Males that were sleep-deprived for 8 h court more than males that were sleep-deprived for 16 h, although they received the same amount of mechanical perturbation. $n = 24$ for each. $***p < 0.001$, one-way ANOVA. Error bars indicate SEM. Please see Supplementary Fig. 1 for the effect of SD on female sexual behavior

also has a role in the maintenance of baseline sleep. To assess this, we used the same intersectional strategy to express tetanus toxin light chain (*TNT*) to block synaptic transmission from P1 neurons (*LexAop2-FlpL/R71G01-LexA; UAS>stop>TNT/dsx^GAL4*). Male flies expressing active *TNT* sleep significantly more as compared to control males expressing an inactive version of *TNT* (*TNT^in*; Fig. 2b). To induce inhibition of P1 neurons conditionally, we also measured sleep in male flies expressing Shibire^ts1 (temperature-sensitive mutation of the *Drosophila* gene encoding a Dynaminorthologue)[43] driven by the *P1-splitGAL4* at 29.5 and 21.5 °C. Control flies showed a decrease in sleep at 29.5 °C, which was suppressed in flies expressing Shibire^ts1 in P1 neurons (Fig. 2g). These results show that inhibition of P1 neurons promotes sleep. Together, these results clearly indicate that activation of sex-promoting P1 neurons suppresses sleep in male flies, while inhibition of P1 neurons increases sleep in males.

It was recently reported that two subsets of *dsx*-expressing neurons (pC1 and pCd, Fig. 2a) promote female receptivity[44]. To determine whether, like in males, sex-promoting neurons influence sleep in females, we assayed sleep in female flies with altered activity of pC1 and/or pCd neurons. We found that females sleep modestly more when pC1 neurons, but not pCd, were activated (Fig. 2d). Activating pC1 and pCd neurons together also increased female sleep by a small amount. However, blocking synaptic transmission from pC1 or pCd neurons did not affect female sleep. These results suggest that, unlike males, increase in female receptivity induced by pC1 and pCd activation does not suppress sleep.

**Activity-dependent regulation of courtship and sleep by P1.** To further study how P1 neurons regulate courtship and sleep in males, we used different temperatures (25.5, 27, 28.5, and 30 °C) to obtain differential activation of P1 neurons, and assay male courtship (wing extension in solitary males) and sleep. Interestingly, we found that P1 activation driven by the *splitGAL4* at 27, 28.5, and 30 °C inhibits male sleep at similar levels (Fig. 3a–d), while activation at 28.5 and 30 °C, but not 27 °C, induces wing extension (Fig. 3i–k). We used the other P1 intersectional driver (*LexAop2-FlpL/R71G01-LexA; UAS>stop> dTrpA1/dsx^GAL4*) and found the same phenotype, except that P1 activation at 27, 28.5, and 30 °C inhibits male sleep at different levels (Fig. 3a–d), which may be due to different populations and/or numbers of P1 neurons targeted by these two methods. These results clearly demonstrate that a lower level of P1 activation is sufficient to inhibit sleep, but a higher level of P1 activation is required for courtship promotion.

As the locomotor activity measured in the beam assays used for sleep analysis is not very sensitive to changes in velocity, we analyzed the locomotor activity of single flies during P1 activation using video recordings over 24 h. We found that P1 activation using the above two drivers at 27 °C slightly increases velocity by ~50% (Fig. 3m), indicative of sleep inhibition, while P1 activation at 28.5 and 30 °C dramatically increases velocity by over five times (Fig. 3m), indicative of increased courtship drive.

**ACh release by P1 neurons is required for sleep regulation.** The above results indicate that P1 neurons inhibit male sleep, but we

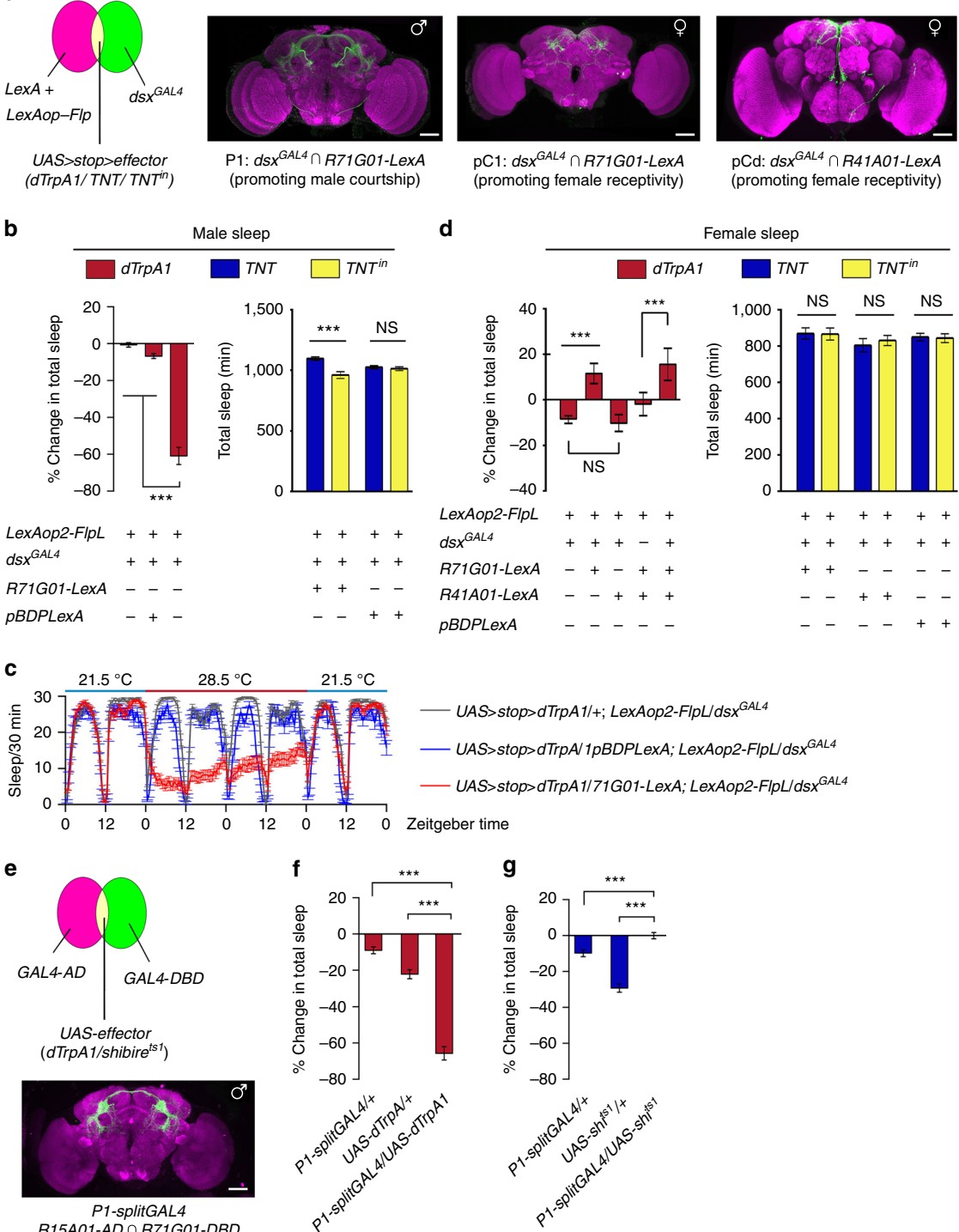

**Fig. 2** Sex-promoting neurons regulate sleep. **a** Diagram of intersectional strategy to label and manipulate subsets of *dsx* neurons: P1 neurons in males promoting male courtship, pC1 and pCd neurons in females promoting female receptivity. **b** Activating P1 neurons suppresses total sleep ($n = 60$), and inhibiting P1 neurons promotes total sleep in male flies. **c** Detailed sleep plot of activating P1 neurons (in *red line*, controls in *blue* and *gray*) in male flies. **d** Activating pC1 but not pCd neurons promotes female sleep, but inhibiting pC1 or pCd neurons does not affect female sleep. **e** Another way of targeting P1 neurons using the *splitGAL4* system. **f** Activation of P1 neurons inhibits male sleep. **g** Silencing P1 neurons slightly increases male sleep compared to controls. $n = 24$-32 for each. ***$p < 0.001$, unpaired *t*-test. *NS*, not significant. Error bars indicate SEM. Scale bars, 50 μm

still do not know the neurotransmitter that P1 neurons release to regulate sleep. Thus, we selectively knocked down neurotransmitters using RNA interference (RNAi) in P1 neurons while activating these neurons via dTRPA1. Activating P1 neurons alone (*UAS-dTrpA1/+; P1-splitGAL4/+*) severely reduced male sleep, but independently knocking down three genes (*Nsf2*, *Syx8*, and *unc-13*), which are known to be required for neural transmission, in P1 neurons fully restores male sleep (Fig. 4a). Furthermore, knocking down acetylcholine (*VAChT* and *Ace*), but not serotonin (*Trh*), dopamine (*DAT* and *Ddc*), octopamine

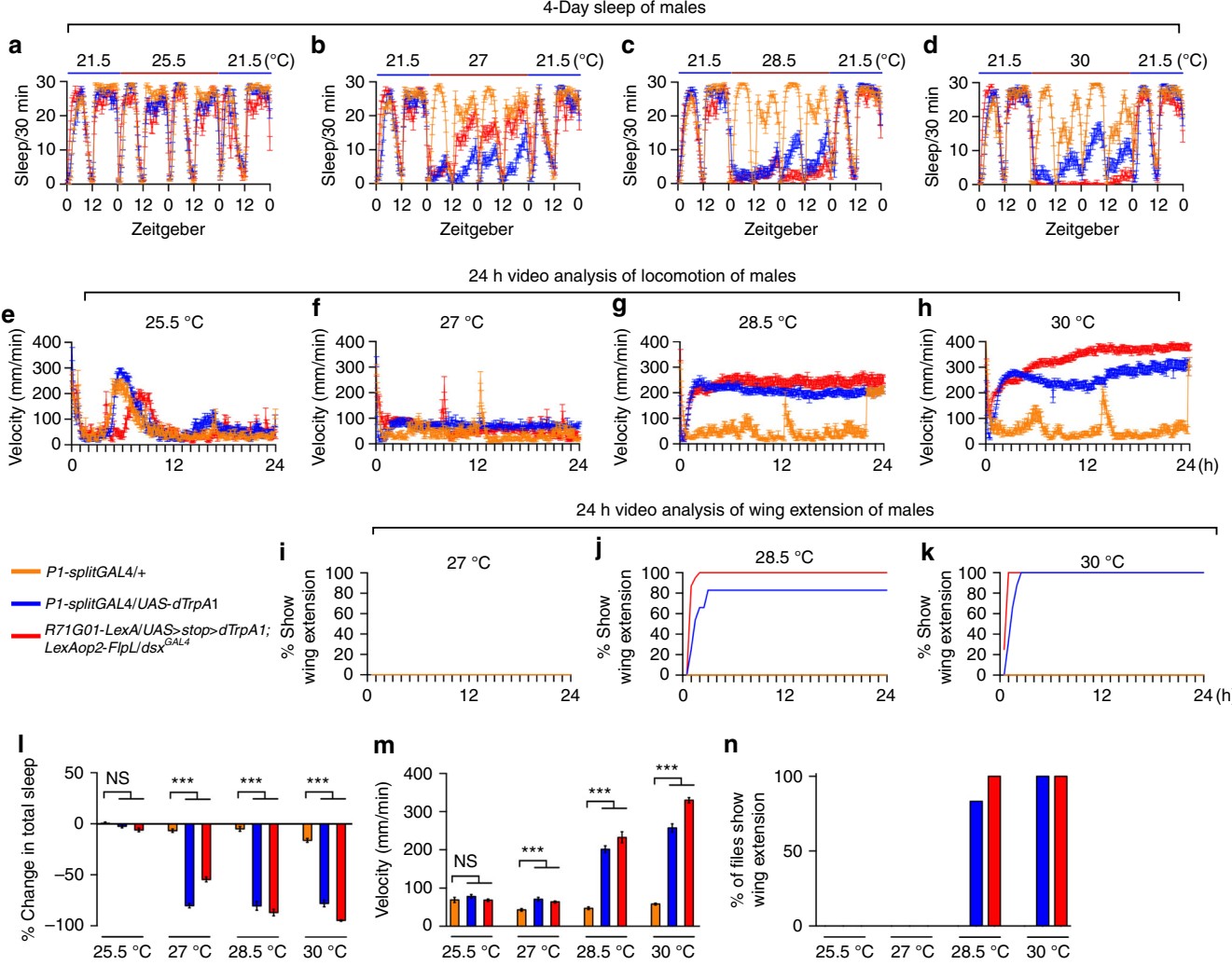

**Fig. 3** Activity-dependent regulation of courtship and sleep by P1. **a–d** Sleep profiles of males with P1 neurons activated at indicated temperatures, as well as control males. $n = 32$ for each. **e–h** Velocity of males with P1 neurons activated at indicated temperatures over 24 h of video tracking. $n = 24$ for each. **i–k** Percentage of males with P1 neurons activated at indicated temperatures shows wing extension over 24 h of video tracking. $n = 48$ for each. **l** Percentage of sleep changes from 21.5 °C to indicated temperatures. **m** The mean velocity of males with P1 neurons activated at indicated temperatures over 24 h. **n** Percentage of males with P1 neurons activated at indicated temperatures show wing extension. ***$p < 0.001$, unpaired $t$-test. NS, not significant. Error bars indicate SEM

(Tdc2), glutamate (VGlut), or GABA (Gad1, Gat, and VGAT), fully restores sleep in P1-activated male flies (Fig. 4a, b). We tested the efficacy of the RNAi reagents targeting the cholinergic system by measuring relative mRNA levels in flies expressing Ace- and VAChT-RNAi using ubiquitous promoter Act5C-Gal4 (Supplementary Fig. 4). Intersectional labeling of $fru^{LexA}$ and neurotransmitter GAL4 lines (cha-GAL4 for acetylcholine, Trh-GAL4 for serotonin, ple-GAL4 for dopamine, Tdc2-GAL4 for octopamine, and dVGAT-GAL4 for GABA, Supplementary Fig. 5) indicates that the male-specific P1 neurons are indeed acetylcholine-positive (Fig. 4c, d), consistent with a previous study using antibody (anti-Cha) staining of dsx-expressing neurons[44]. Thus, we conclude that acetylcholine release from P1 neurons is required for sleep regulation in males, although the direct synaptic target of P1 neurons that mediates sleep in males is still unknown.

**P1 neurons regulate sleep through $fru^M$-positive DN1 neurons.** We hypothesized that the activation of P1 neurons might function through known sleep circuitry to suppress sleep. To test

this hypothesis, we performed "neuronal epistasis" experiments in which we simultaneously activated P1 neurons with dTRPA1 and blocked synaptic outputs of candidate downstream sleep-regulating neurons with Shibire[ts1] (Fig. 5a, b). We selected a set of LexA driver lines (Supplementary Fig. 6) targeting candidate neurons that have been shown to be involved in sleep: mushroom body Kenyon cells (R14H06-LexA, R35B12-LexA, and R44E04-LexA), mushroom body output neurons (R12C11-LexA, R14C08-LexA, R24H08-LexA, R25D01-LexA, and R71D08-LexA), PAM dopaminergic neurons (R58E02-LexA), fan-shaped body neurons (R23E10-LexA and R84C10-LexA), and DN1 circadian clock neurons (R18H11-LexA). Blocking synaptic outputs of any of these neurons with Shibire[ts1] does not significantly affect sleep (Fig. 5b), although activating many of these neurons affects sleep[26, 27]. This could be a result of basal activity of these neurons already being relatively low, as has been observed for other sleep-regulating neurons[33], or differences in strength of expression of driver lines.

On the basis of this screen of putative downstream effectors of P1 neurons, we find that silencing specifically the synaptic

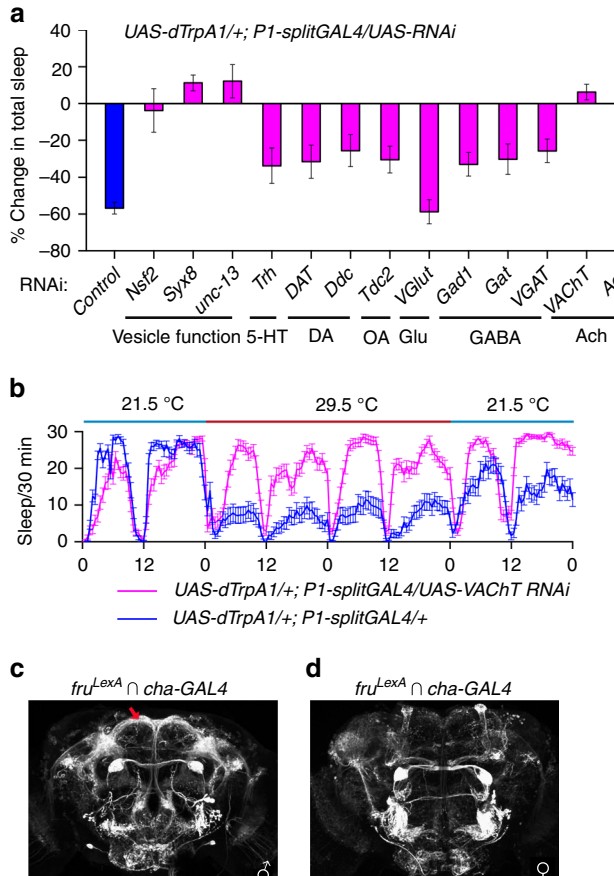

**Fig. 4** ACh release by P1 neurons is required for sleep regulation. **a** Knocking down Ach in P1 neurons abolishes the effect of P1 activation on male sleep. The *blue bar* indicates P1 activation alone, while *bars in magenta* indicate P1 activation together with knocking down specific genes in P1 neurons. $n = 32$ for each. **b** Detailed sleep profiles of P1 activation alone (in *blue line*) and P1 activation together with VAChT knocked down (in *magenta line*). **c, d** Intersectional expression of $fru^{LexA}$ and *cha-GAL4* in a male brain **c**, where P1 neurons are labeled, and in a female brain **d**. Error bars indicate SEM. Scale bars, 50 μm

outputs of a subset of dorsal clock neurons: DN1 neurons partially reverses the sleep deficit induced by activating P1 neurons (Fig. 5d), suggesting that P1 neurons suppress sleep at least in part by activating DN1 sleep-regulating circadian clock neurons. These sleep changes are not due to differences of general locomotor activity between the permissive (21.5 °C) and restrictive (29.5 °C) temperatures (Supplementary Fig. 7).

DN1 neurons express FRU$^M$ and regulate courtship activity rhythms[31]. Furthermore, DN1 neurons are known to suppress sleep through secretion of the wake-promoting neuropeptide diuretic hormone 31 (Dh31)[27]. Functional imaging of DN1 neurons using genetically encoded fluorescent voltage indicator reveals that the Dh31-expressing DN1s are electrically active before dawn (ZT 22) as compared to late in the day (ZT10), supporting the hypothesis that DN1 activity awakens the fly[27]. We also tested whether the P1-DN1 circuit mediates courtship by activating P1 neurons and inhibiting DN1 neurons as described above and found that, unlike sleep, P1 activation-induced courtship behaviors were not suppressed by DN1 inhibition (Supplementary Fig. 8).

Following up on the neuronal epistasis experimental results and a role for P1-DN1 circuit in sleep suppression, we sought to

determine whether the DN1 clock neurons are synaptically downstream of P1 neurons. Double-labeling (data not shown) and brain registration (Fig. 5e and Supplementary Movie 1) of DN1 and P1 neurons suggest that DN1 neurons are not directly connected with P1 neurons. Indeed, there is no GRASP (GFP reconstitution across synaptic partners)[45] signal between DN1 and P1 neurons (Fig. 5f), suggesting indirect polysynaptic connection or no connection between P1 and DN1 neurons. To test the possibility of indirect poly-synaptic connectivity, we transiently activated P1 neurons and recorded Ca$^{2+}$ signals from DN1 neurons. We found that activation of P1 neurons expressing ATP-dependent depolarizing ion channel P2X$_2$[46] with a puff of 10 mM ATP induced robust calcium responses in the cell bodies and projections of DN1 neurons (R18H11-LexA/LexAop2-GCamp6m; R15A01-GAL4/UAS-P2X2; Fig. 6a–c and Supplementary Movie 2). DN1 calcium responses were not observed in brains that lack P2X$_2$ expression in P1 neurons (R18H11-LexA/LexAop2-GCamp6m; R15A01-GAL4/+, Supplementary Movie 3). These results demonstrate that DN1 neurons are indeed synaptically downstream of P1 neurons and, based on the above anatomical evidence, this synaptic connectivity is indirect.

We tested whether the DN1 neurons feedback into the P1 neuronal cluster and found that activating the DN1 neurons activates the P1 neurons measured by an increase in Ca$^{2+}$ response (Fig. 6d–f and Supplementary Movie 4), which may be a mechanism for rhythmic control of courtship by DN1 neurons reported previously[31], although we found that activating DN1 neurons via dTRPA1 does not significantly change courtship (Supplementary Fig. 8). These data provide direct evidence for mutually excitatory interactions between DN1 and P1 neurons that support a positive feedback circuit model that likely underlies persistence of arousal states associated with sleep-suppression and courtship. Thus, the reciprocal interactions between P1 and DN1 neurons link the sleep and courtship drive and are critical to the behavioral choice. On the basis of the above results we propose that activity of P1 neurons is directly influenced by the sleep-controlling DN1 neurons and that P1 activity is central to the sleep–sex behavioral switch.

**P1 activity is suppressed by SD**. To test whether the activity of P1 neurons is directly modulated by sleep need, we directly measured spontaneous neural activity of these neurons in sleep-deprived and sleep-replete flies as described[33]. We expressed the genetically encoded fluorescent voltage indicator *ArcLight*[47] using the above *P1-splitGAL4* driver, and imaged spontaneous activity of the lateral junction region of P1 neurons in males (Fig. 7a). We found that P1 activity is significantly reduced in SD males as compared to sleep-replete controls that were imaged in parallel (Fig. 7b, c). We also analyzed these results using comparisons between peak or maximal $\Delta F/F_0$ and found significant differences (Supplementary Fig. 9). Furthermore, wing extension by P1-activated males is significantly reduced by SD (Supplementary Fig. 10). These data strongly support the observation that SD males have reduced courtship possibly resulting from diminished activity of the P1 courtship command neurons. Furthermore, activating P1 neurons with dTRPA1 in SD males restores male courtship (Fig. 7d). These data further support the hypothesis that reduced activity of DN1 neurons[27] in sleepy males reduces excitatory input into P1 neurons, thereby preventing the flies from engaging in wake-associated social behaviors. As P1 neurons receive inputs from multiple sensory inputs, it is likely that sleep deprivation also modulates activity of non-DN1 inputs into the P1 neurons. Thus, P1 neurons regulate

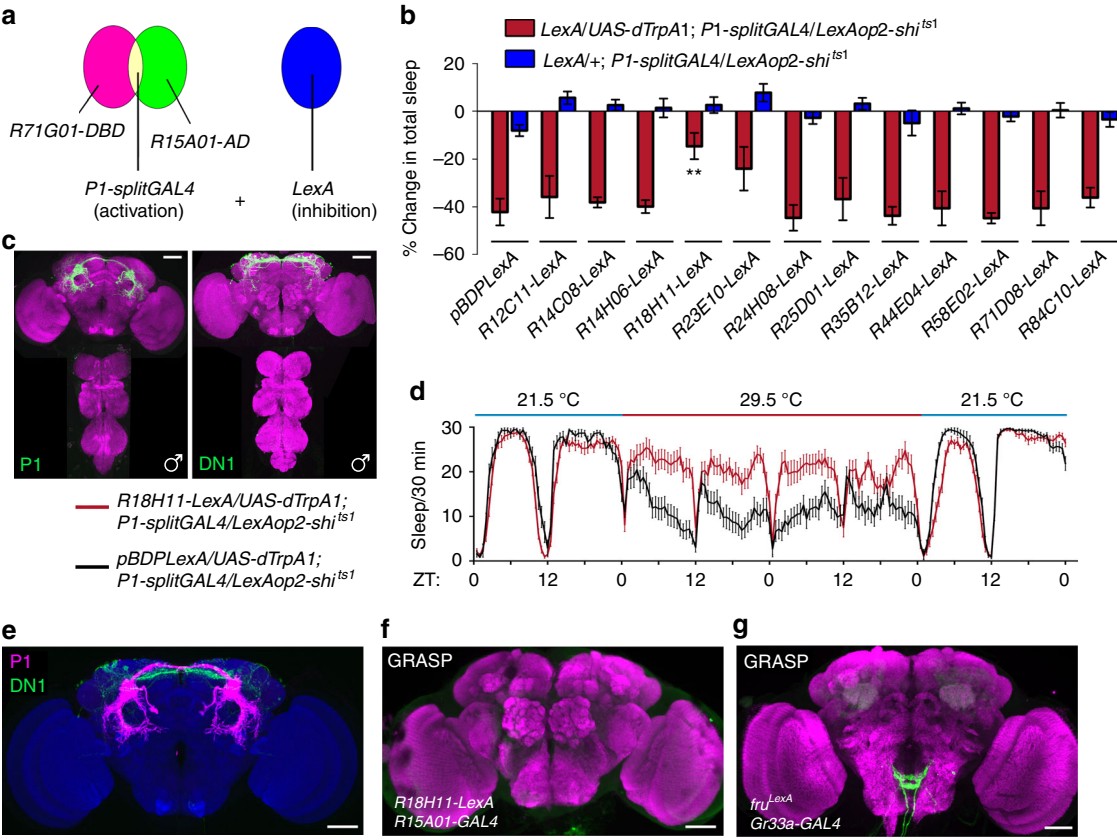

**Fig. 5** P1 regulates male sleep through DN1 neurons. **a** Diagram for activating P1 neurons in the sex circuitry, and inhibiting subsets of neurons in the sleep circuitry. **b** Identification of *R18H11-LexA* DN1 circadian clock neurons as functionally downstream of P1 neurons. *Blue bars* indicate *LexA* inhibition alone, while *red bars* indicate P1 activation together with *LexA* inhibition. *n* = 24–56 for each. \*\**p* < 0.01, Kruskal–Wallis non-parametric one-way ANOVA followed by *post hoc* Dunn's correction. *Error bars* indicate SEM. **c** Expression pattern of *P1-splitGAL4* and *R18H11-LexA* driving *UAS-myrGFP* and *LexAop-myrGFP*, respectively. **d** Detailed sleep profiles of P1 activation alone (in *black line*) and P1 activation together with DN1 inhibition (in *red line*). **e** Brain registration of P1 (*magenta*) and DN1 (*green*) neurons. **f** There is no GRASP signal between P1 and DN1 neurons. **g** GRASP signals between Gr33a-expressing neurons and *fru^M*-expressing neurons as a positive control. Scale bars, 50 μm

male courtship by integrating external courtship stimuli and internal sleep needs.

**Sex-specific control of sleep by *fruitless* and *doublesex*.** The above results indicate that selection of sleep and sexual behaviors is sexually dimorphic, but we also note that the sleep or sexual behaviors *per se* are sexually dimorphic. Further, P1 and DN1 neurons identified in regulation of the behavioral choice between sleep and courtship express sex-specific genes *fru^M* and/or *dsx*. To elaborate the role of these genes in behavioral choice, we sought to understand the role of sexually dimorphic genes in individual behaviors. Although it has been well studied how *fru^M* and *dsx* control sex-specific sexual behaviors, whether they also control sex-specific sleep is unclear.

Male flies sleep more than females[28, 29]. We investigated whether such sexually dimorphic sleep amounts of male and female flies are controlled by the sex determination master gene *transformer* (*tra*), and its two downstream target genes *fru^M* and *dsx*[48, 49] (Fig. 8a). Knockdown of *tra* in the nervous system using pan-neuronal *c155-GAL4* to drive expression of RNAi from *UAS-traIR* increases female sleep to match that of male flies, while parental control females (*c155/+* and *UAS-traIR/+*) sleep significantly less than males of the same genotype (Fig. 8b, c). This indicates that sexually dimorphic sleep quantity is indeed controlled by the sex determination pathway.

We then tested the role of the *dsx* and *fru^M* branches in sleep regulation. We found that females expressing RNAi targeting *dsx* pan-neuronally (*c155>UAS-dsxIR*) sleep even slightly more than males of the same genotype, while parental control females sleep less than males (Fig. 8b, c). We also found that females pan-neuronally expressing microRNAs targeting *fru^M*[50] (*c155>UAS-fruMi*) sleep as much as males of the same genotype, while control females expressing a scrambled version (*UAS-fruMiScr*) sleep less than males (Fig. 8b, c). These results further support the role of sex determination genes in regulating sleep.

To further investigate how *dsx* and *fru^M* regulate sexually dimorphic sleep, we tested multiple combinations of *dsx* and *fru^M* alleles. We found that for two *dsx* null genotypes (*dsx^683–7058/dsx^1649–9625* and *dsx^683–7058/dsx^M+R15*), and a masculinized line (*dsx^683–7058/dsx^M*) that expresses *dsx^M* regardless of sex, the amounts of male sleep are similar to parental control males (Fig. 8d); but the amounts of female sleep are significantly greater than in parental control females (Fig. 8e). Furthermore, the increment of sleep in *dsx* null females is specific to daytime sleep, which contributes to the majority of sexual dimorphic sleep (Supplementary Fig. 11). In contrast, for all five *fru^M* null genotypes we tested, the amounts of male sleep are significantly lower than that in parental control males (Fig. 8f), while the amounts of female sleep are similar to what is observed in parental control females (Fig. 8g). The decrement of sleep in *fru^M* null males is not specific to daytime or nighttime (Supplementary

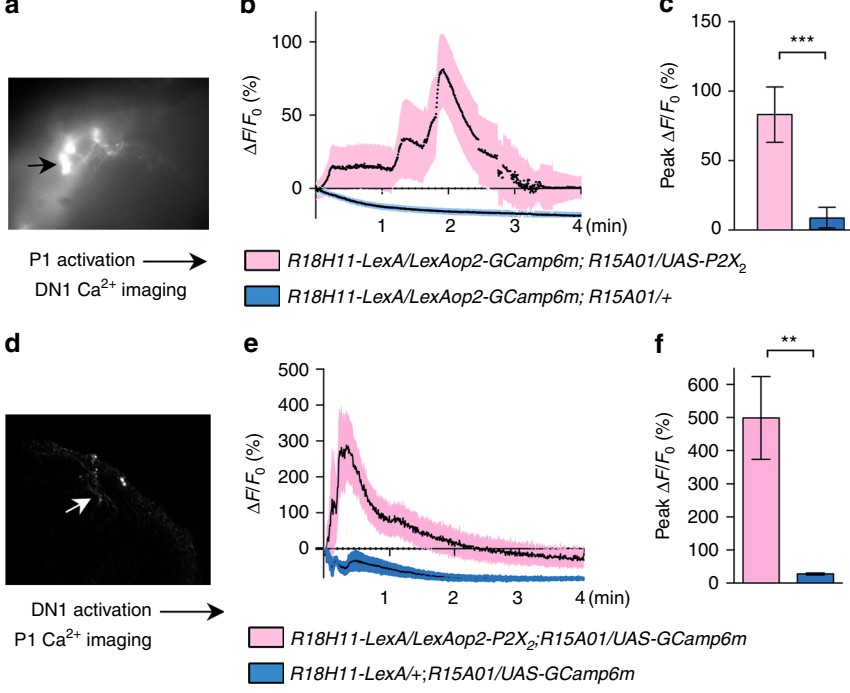

**Fig. 6** P1 and DN1 neurons form mutually excitatory connections. **a** *Movie still* of Calcium imaging of DN1 cell bodies after P1 activation via P2X$_2$. **b** Averaged traces of $\Delta F/F_0$ of DN1 neurons after P1 activation. **c** Peak fluorescence changes ($\Delta F/F_0$) of DN1 neurons after P1 activation. $n = 6$ for each, ***$p < 0.001$. Unpaired *t*-test. **d** *Movie still* of calcium imaging of P1 neurites in the lateral protocerebrum region after DN1 activation via P2X$_2$. **e** Averaged traces of $\Delta F/F_0$ of P1 neurons after DN1 activation. **f** Peak fluorescence changes ($\Delta F/F_0$) of P1 neurons after DN1 activation. $n = 10$ for each, **$p < 0.01$. Unpaired *t*-test. Error bars indicate SEM

Fig. 12). Taken together, we conclude that DSX$^F$ in females inhibits specifically daytime sleep, while FRU$^M$ in males promote sleep during both daytime and nighttime. However, FRU$^M$ is not sufficient to affect sleep in the presence of DSX$^F$, as females expressing ectopic FRU$^M$ (and have DSX$^F$) sleep similarly to control females.

**FRU$^M$ differentially regulates courtship and sleep.** It has been proposed that FRU$^M$ specifies a neuronal circuitry that is dedicated for male courtship behavior[13, 14]. As we found that FRU$^M$ also regulates male sleep, we set out to identify the neuronal substrates where FRU$^M$ functions to mediate sleep. We used the microRNA, as mentioned above, to target *fru*$^M$ in subsets of *fru*$^M$ neurons, and a scrambled version as control. Knockdown of FRU$^M$ in all *fru*$^M$ neurons (*fru*$^{GAL4}$) significantly decreases male sleep, but driving the microRNA in glia cells (*repo-GAL4*), or a subset of P1 neurons (*R71G01* and *R15A01*), that are crucial for male courtship, did not affect male sleep (Fig. 9a, b). Knockdown of FRU$^M$ in all *dsx* neurons (*dsx*$^{GAL4}$) only slightly altered male sleep (reduced by ~6%, Fig. 9a, c). However, we observed significant decrement of male sleep when knocking down FRU$^M$ in MB Kenyon cells (*R13F02*, *R19F03*, and *R76D11*, Fig. 9a, d, e), or DN1 neurons (*R18H11*, Fig. 9a, f), all of which express FRU$^M$ (Supplementary Fig. 13). These sleep changes are not due to general locomotor activity differences as indicated by activity during the wake phase (Supplementary Fig. 14).

On the basis of our neuronal epistasis and functional imaging experiments, the DN1 neurons seem critical to the male-specific sex–sleep selection mechanism. We investigated the precise mechanism by which FRU$^M$ regulates sleep in the DN1 neurons. We recently found that the neuropeptide Dh31 functions in DN1 neurons to regulate fly sleep[27], thus we asked whether

FRU$^M$ might regulate sleep through Dh31 in DN1 neurons. We tested sleep in males with FRU$^M$ knocked down in DN1 neurons (*R18H11/UAS-fruMi*) in the background of Dh31 mutants. We found that FRU$^M$-mediated sleep loss is dependent on Dh31, as a single copy of a Dh31 allele (*Dh31*$^{KG09001}$) already attenuates the sleep loss, and a combination of Dh31 alleles (*Dh31*$^{KG09001}$/*Df(Dh31)*) almost abolishes the sleep loss (Fig. 9a, g). Thus, FRU$^M$ promotes male sleep in DN1 neurons by regulating Dh31 levels and/or secretion.

To investigate how FRU$^M$ regulates both male courtship and sleep, we then tested courtship behavior of the above FRU$^M$ knocked down males, and found that knocking down FRU$^M$ in *dsx*-expressing neurons severely impairs male courtship, but knocking down FRU$^M$ in MB neurons or Dh31-expressing DN1 neurons does not affect male courtship (Fig. 9h). Thus, FRU$^M$ functions in distinct neural substrates to regulate male courtship (e.g., *fru*$^M$ and *dsx* overlapping P1 neurons) and male sleep (e.g., MB and DN1 neurons).

**Discussion**

Neural networks integrate external sensory cues, past experience, and internal states to control key behavioral decision-making. How these neural networks support behavioral choice critical for reproduction and survival at both the individual and species level is poorly understood. Here we focused our attention on identifying and characterizing the molecular and neural basis of reciprocal control of sleep and reproductive behaviors. The neuronal mechanism we have uncovered involves the P1 neurons implicated in courtship decision-making and the DN1 neurons, a part of the core clock and sleep circuit in suppressing sleep. Recently, the activity of P1 neurons was shown to be modulated by excitatory and inhibitory inputs from gustatory and olfactory

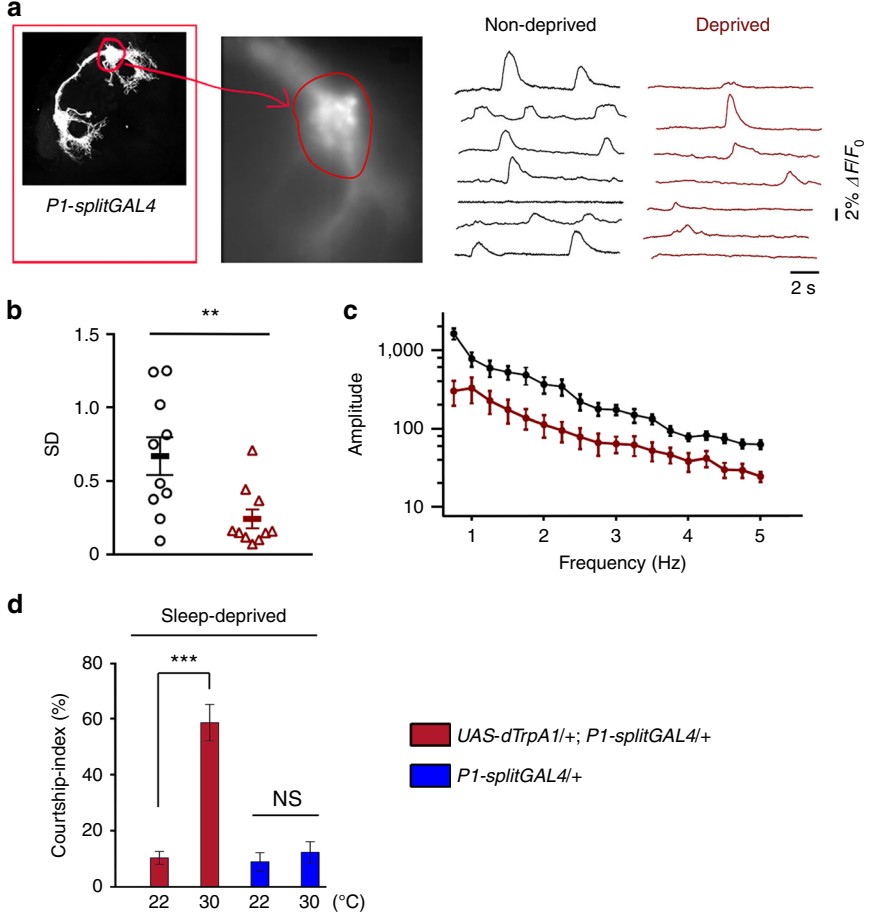

**Fig. 7** SD decreases P1 activity. **a** Optical recording of membrane activity in the lateral junction (*circled*) of P1 neurons in control (non-deprived) males and sleep-deprived males. **b** SD of the optical signal was plotted, with mean ± SEM. $n = 10$ for each. **$p < 0.01$, unpaired *t*-test. **c** Power spectrum was computed using fast Fourier transform with 0.2 Hz bin width. Power of the non-deprived group is significantly greater than the sleep-deprived group (two-way ANOVA with repeated measures). **d** Activating P1 neurons with dTRPA1 overcomes courtship suppression by SD. $n = 24$ for each. ***$p < 0.001$, unpaired *t*-test. *NS*, not significant. Error bars indicate SEM

systems and thus processing multisensory information controlling behavioral output relevant to male courtship[39, 40]. In addition to the gustatory and olfactory input into the P1 neurons, a subset of dopaminergic neurons in the anterior superior medial protocerebrum has also been implicated in modulating P1 activity[51]. Interestingly, the dopamine activity and its influence on P1 neurons are dependent on mating history. P1 activity is also modulated by housing conditions (e.g., single-housed vs. group-housed)[38].

The male-specific P1 neurons function as the key courtship command neurons that trigger a spatial and temporal pattern of motor neuron activity specific to courtship behaviors[35–37]. We find that this higher-order processing within the P1 neurons is altered as a result of sleep and wake states as evidenced by decreases in spontaneous activity of the P1 neurons after SD. We also found that P1 neurons could be activated by sleep-controlling DN1 neurons. Additionally, activating P1 neurons (gain of function) decreases sleep while inhibiting P1 neurons (loss of function) increases sleep. These data show that P1 neurons represent an important sleep-regulating locus in the male fly brain.

While the neural circuitry downstream of P1 neurons important in producing courtship behaviors is well understood, the circuit mechanisms by which P1 neurons produce wake behavior are intriguing. Using P2X$_2$ to activate P1 neurons and GCamp6m

to visualize DN1 activity, we found that DN1 neurons are functionally downstream of P1 neurons, although they do not have such a direct synaptic connection. To ensure that this specific connection underlies the wake-promoting phenotype of P1 neuron activation, we simultaneously activated the P1 neurons while inhibiting the DN1 clock neurons and observed a strong decline in the wake-promoting phenotype of P1 activation. This effect was not phenocopied by silencing other key sleep-regulating centers of the fly brain including mushroom body and central complex neurons.

Our results demonstrate that P1 and DN1 neurons form a positive feedback loop and support persistent neural activity to sustain extended phases of arousal necessary for social behaviors like courtship. The role of P1-DN1 mutual excitatory circuit as the mechanistic basis of interaction between sleep and courtship drive is further supported by the finding that P1 activity is low in sleepy flies as compared to sleep-replete controls. The P1 neural node activity is highly dynamic and is influenced by sensory inputs and social experiences, which could further underlie the reduced neural activity of P1 neurons in sleep-deprived flies. It is also interesting to note that our induced activation studies of P1 neurons using dTRPA1 show that low levels of activation are sufficient to suppress sleep, but a higher level of activation is required for courtship promotion.

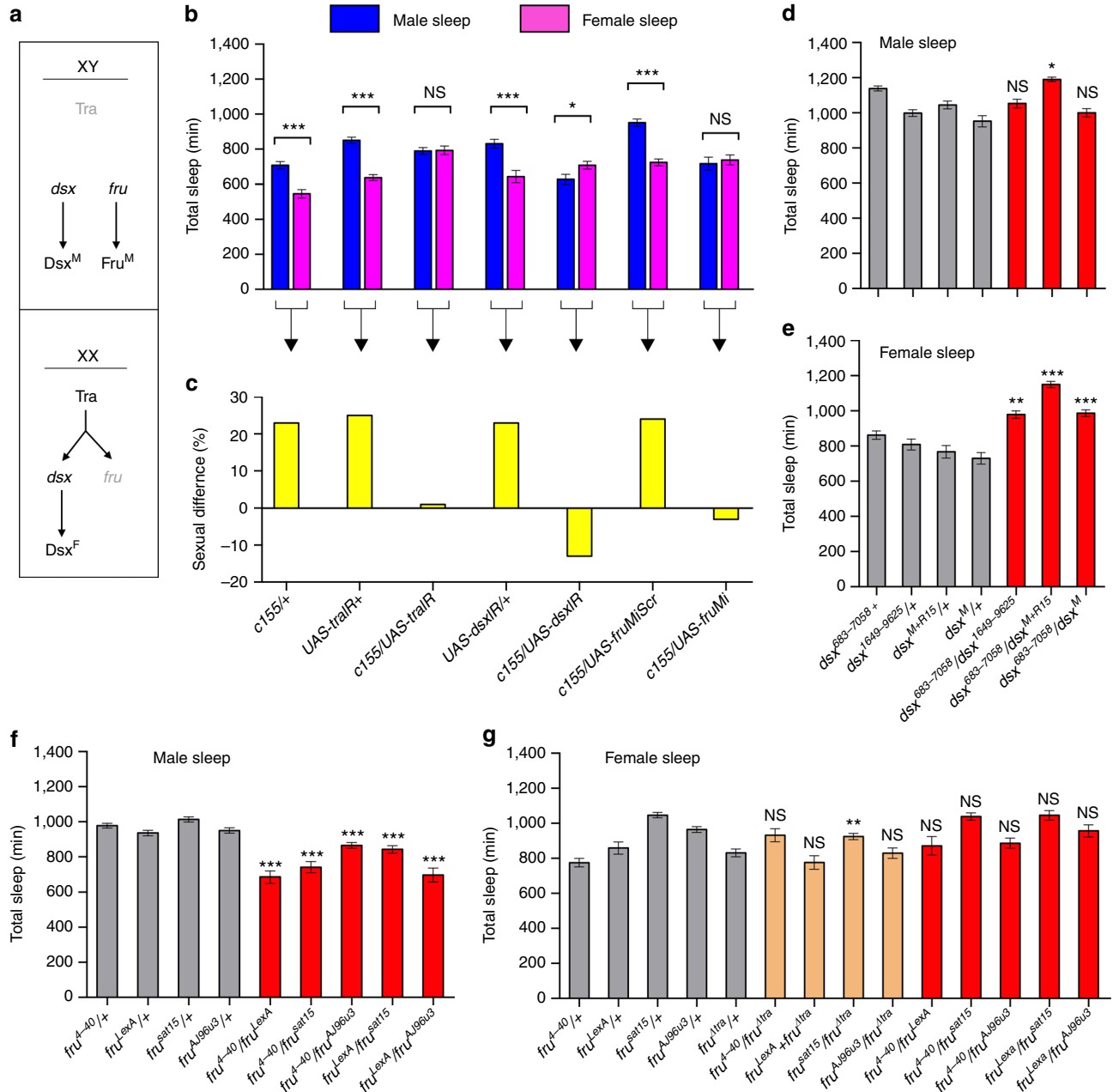

**Fig. 8** Sex determination genes regulate sleep. **a** Sex determination pathway in *Drosophila*. **b, c** Sex determination genes *tra*, *fru*[M], and *dsx* regulate sexually dimorphic sleep quantity. Total sleep amounts of males and females **b**, and the corresponding sexual difference of sleep **c** are shown for indicated genotypes. n = 24-56 for each. *p < 0.05 and ***p < 0.001, unpaired *t*-test. NS, not significant. **d-g** Total sleep in males (**d**) and females (**e**) of *dsx* alleles, and total sleep in males (**f**) and females (**g**) of *fru*[M] alleles. n = 24-32 for each. *p < 0.05, **p < 0.01, and ***p < 0.001; comparisons are made between the genotype and its parental genotypes, one-way ANOVA. NS, not significant. Error bars indicate SEM

At the molecular level, we find that acetylcholine release from the P1 neurons is critical to sleep regulation such that inhibiting synthesis or release of Ach from the P1 neurons suppresses the wake behavior induced by P1 activation.

Interestingly, the reciprocal regulation of sleep and sexual behaviors is different in male and female flies, leading to sex-specific interaction of these behaviors. Note that P1 neurons express FRU[M] and DSX[M], and are male-specific, while DN1 neurons also express FRU[M]. Recent studies have also shown that the DN1 neurons are more active in males as compared to females, further supporting our key findings of male-specific courtship–sleep circuitry involving P1-DN1 neuronal connections[52].

The male-specific courtship–sleep interaction mediated by the P1-DN1 circuit is not surprising as both sexual and sleep behaviors per se are sexually dimorphic. Although it has been well documented that male and female flies have different sleep patterns and that female flies sleep less during the daytime as compared to males, the molecular and neural basis of this sexual dimorphism in sleep patterns is unclear. Sex differences in nervous system structure and behaviors are all attributed to sex determination genes *tra*, *fru*[M], and *dsx*[48, 49]. We found that these genes also control sexually dimorphic sleep patterns. In particular, FRU[M] and DSX[F] differentially regulate male and female sleep, respectively, where FRU[M] promotes male sleep during both daytime and nighttime, and DSX[F] inhibits female

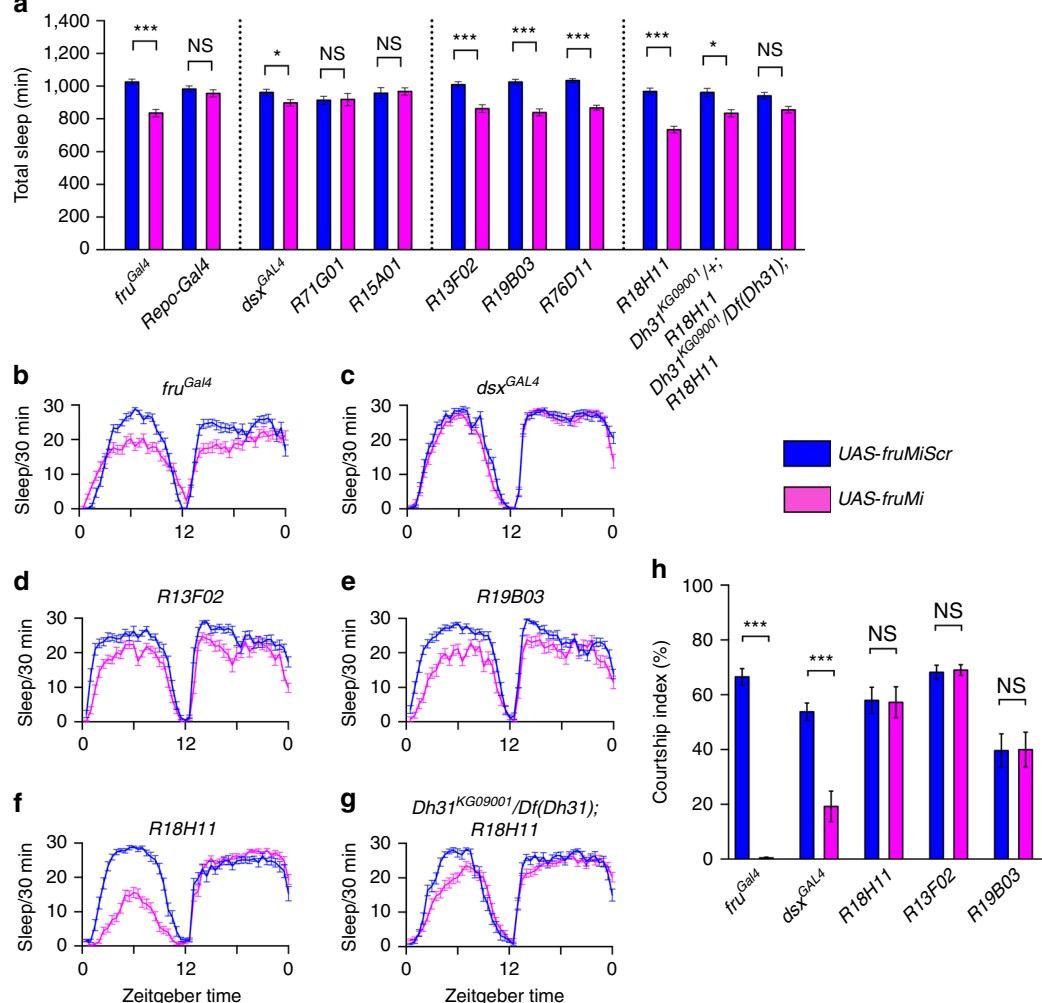

**Fig. 9** FRU[M] differentially regulates courtship and sleep. **a** Knocking down *fru[M]* in MB or DN1 neurons, but not P1 neurons, reduces male sleep, and the function of *fru[M]* in DN1 neurons on male sleep is dependent on Dh31. **b–g** Detailed sleep profile of males with *fru[M]* knocked down in all *fru[M]*-expressing neurons (**b**), all *dsx*-expressing neurons (**c**), MB neurons (**d**, **e**), DN1 neurons (**f**), and DN1 neurons in Dh31-mutated background (**g**). **h** Knocking down *fru[M]* in *dsx*-expressing neurons but not MB or DN1 neurons impairs male courtship. $n = 24\sim32$ for each. $*p < 0.05$ and $***p < 0.001$, unpaired *t*-test. NS, not significant

sleep during daytime but not nighttime. Furthermore, we found that knockdown of FRU[M] in MB or DN1 neurons decreases male sleep amount, but leaves male courtship intact; in contrast, knockdown of FRU[M] in all *dsx*-expressing neurons severely impairs male courtship, but only slightly alters sleep, suggesting that FRU[M] functions in distinct neural substrates to regulate male courtship and sleep.

In order to understand the complex relationship between the male-specific P1-DN1 feedback and sexually dimorphic sleep patterns, we probed how FRU[M] acts on the DN1 neurons in regulating male sleep. DN1 neurons have been previously implicated in sleep regulation and release a wake-inducing neuropeptide Dh31. Here we find evidence that FRU[M] modulates the production or secretion of Dh31 specifically from the DN1 neurons to regulate sleep. The neural circuit mechanism by which Dh31 released from DN1 neurons regulates wake is unknown, but there is evidence of synaptic communication between the DN1 and Pars intercerebralis neurons, another sleep-regulating loci[53]. Further, the pars intercerebralis neurons regulate circadian output by release of another neuropeptide Dh44[53, 54].

Previous studies have looked at more generalized decision-making neurons, but here we have identified a novel sex-specific neuronal circuitry for sex–sleep interaction, which depends on sex determination genes that directly influence the neuronal output of key decision-making neurons (Fig. 10). Thus, our findings on genetic and neural circuit mechanisms underlying sex–sleep interaction will have broad implications for studies on decision-making and behavioral choice in higher-order organisms.

## Methods

**Fly stocks.** Flies were maintained at 22 or 25 °C in a 12 h:12 h light:dark cycle. *fru[M]* alleles used in this study include *fru[LexA]*, *fru[4−40]*, *fru[AJ96u3]*, *fru[sat15]*, and *fru[Δtra]*. *dsx* alleles are *dsx[GAL4(Δ2)]*, *dsx[683−7058]*, *dsx[1649−9625]*, *dsx[M+R15]*, and *dsx[M]*. *Dh31* alleles are *Dh31[KG09001]* and *Df(Dh31)*. RNAi lines are from Tsinghua Fly Center (THFC) at the Tsinghua University[55, 56]. Neurotransmitter *GAL4* lines are from Bloomington Stock Center. *R12C11, R14C08, R14H06, R15A01, R18H11, R23E10, R24H08, R25D01, R35B12, R41A01, R44E04, R58E02, R71D08, R71G01,* and *R84C10* are enhancers for Janelia *GAL4* (or *GAL4-AD, GAL4-DBD*) or *LexA* drivers[57−59]. *LexAop-FlpL, LexAop-shi[ts1]*, and *pBDPGAL4[58], UAS-dTrpA1[60], LexAop-GCamp6m, UAS-GCamp6m[61], UAS-P2X₂[46], UAS-fruMi,* and *UAS-fruMiScr[50], UAS-ArcLight[47], UAS-CD4::spGFP1-10* and *LexAop-CD4::spGFP11[45],*

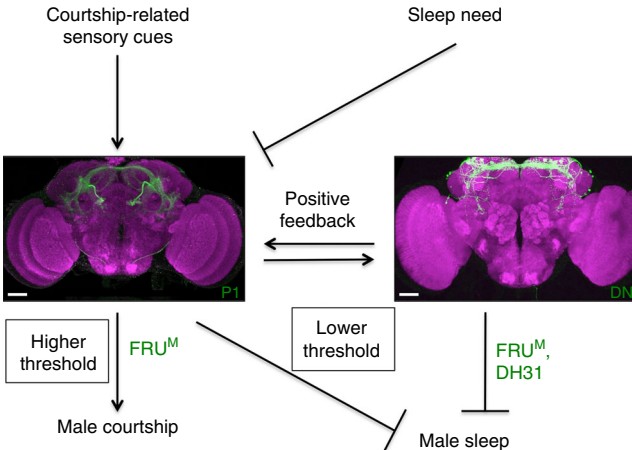

**Fig. 10** Sex-specific interaction between courtship and sleep. The male-specific P1 neurons integrate external courtship-related sensory cues and internal sleep need, then inhibit male sleep, and promote male courtship in a threshold-dependent manner. P1 neurons form mutually excitatory connections with sleep-controlling DN1 neurons, which might be important for a persistent behavioral state. Furthermore, FRU$^M$ differentially regulates male courtship and sleep in P1 and DN1 neurons. Scale bars, 50 µm

UAS>stop>dTrpA1[62], UAS>stop>TNT and UAS>stop>TNT[in14] have been described previously.

**Male courtship assay**. Four to eight-day-old wild-type virgin females were decapitated and loaded individually into round two-layer chambers (diameter: 1 cm; height: 2.5 mm per layer) as courtship targets, and 4–6-day-old tester males were then gently aspirated into the chambers and separated from target females by a plastic transparent barrier until courtship test for 10 min. Courtship tests start roughly 30 min after the end of SD. Courtship index, which is the percentage of observation time a male fly performs courtship, was used to measure courtship to female targets, and measured manually using the LifeSongX software.

**Female receptivity assay**. Wild-type virgin females (4–6 days old) were sleep-deprived and aspirated into round two-layer chambers (diameter: 1 cm; height: 2.5 mm per layer), and separated from 4- to 6-day-old wild-type virgin males until courtship test for 15 min. Courtship tests start roughly 30 min after the end of SD.

**Sleep test and analysis**. Individual 2–4-day-old males or females were placed in locomotor activity monitor tubes (DAM2, TriKinetics Inc.) with fly food, and were entrained in 21.5 or 25 °C 12 h:12 h light:dark conditions for at least 2 days before sleep test. For temperature-shifting experiments, 2-day sleep data were recorded at 21.5 °C as baseline, and then flies were shifted to 28.5 °C (dTrpA1) or 29.5 °C (dTrpA1 and shi[ts1]) for 2 days, and returned to 21.5 °C for at least 1 day to measure sleep recovery. Sleep was analyzed using custom-designed Matlab software[63]. The total sleep amounts per day (e.g., Fig. 8b) are the average sleep amounts in multiple testing days. The sexual difference of sleep (Fig. 8c) is defined as the total sleep of females subtracted from the total sleep of males of the same genotype, and divided by the total sleep of males. Change in total sleep (e.g., Fig. 2f) is the percentage of sleep change in the first day of temperature shift (28.5 or 29.5 °C) compared to baseline sleep at 21.5 °C.

**Sleep deprivation**. SD was performed either in food vials or in DAM2 monitors that are fixed in a multitube vortexer. The vortexer shakes the vials or monitors in an intermittent way (in total ~15 s/min shaking unless specifically described) controlled by the TriKineticsDAMSystem software[33].

**Tissue dissection, staining, and imaging**. Brains and ventral nerve cords of 4–6-day-old males and females were dissected in Schneider's insect medium (S2) and fixed in 2% paraformaldehyde in S2 medium for 50–60 min at room temperature. After 4 × 10-min washing in PAT (0.5% Triton X-100, 0.5% bovine serum albumin in phosphate-buffered saline), tissues were blocked in 3% normal goat serum (NGS) for 90 min, then incubated in primary antibodies diluted in 3% NGS for 12–24 h at 4 °C, then washed in PAT, and incubated in secondary antibodies diluted in 3% NGS for 1–2 days at 4 °C. Tissues were then washed thoroughly in

PAT and mounted for imaging. Antibodies used were rabbit anti-GFP (Invitrogen A11122) 1:1000, mouse anti-Bruchpilot (Developmental Studies Hybridoma Bank nc82) 1:30, and secondary Alexa Fluor 488 and 568 antibodies (1:500). Samples were imaged at ×20 magnification on Zeiss 700 or 710 confocal microscopes and processed with Fiji software.

**Brain image registration**. The standard brain used in this study is described previously[44]. Confocal images for R18H11-LexA and P1-splitGAL4 were registered onto this standard brain with a Fiji graphical user interface as described previously[64, 65].

**P2X$_2$ activation and GCamp6m imaging**. To gain access to the P1 or DN1 neurons for ATP application, whole-brain explants were placed on 8-mm diameter circular coverslips and placed in a recording chamber containing external solution (103 mM NaCl, 3 mM KCl, 5 mM N-tris methyl-2-aminoethane-sulfonic acid, 8 mM trehalose, 10 mM glucose, 26 mM NaHCO$_3$, 1 mM NaH$_2$PO$_4$, 2 mM CaCl$_2$, and 4 mM MgCl$_2$, pH 7.4). For an early experiment (Fig. 6a–c), an ATP ejection electrode was filled with freshly prepared 10 mM ATP solution and positioned near P1 neurons using a micromanipulator. The ATP was ejected by applying a 25-ms pressure pulse at 20 psi using a picospritzer (Parker Hannifin, Precision Fluidics Division, NH). The picospritzer was triggered by Zeiss image acquisition and processing software Zen pro 2012. Later, we added ATP simply using a pipette, but the final ATP concentration is still 10 mM (Fig. 6d–f). Calcium imaging was performed using Zeiss AxoExaminer Z1 upright microscope with W Plan Apochromat 40 × water immersion objective. GCamp6m was excited with a 470-nm LED light source (Colibiri, Zeiss) and images were acquired using ORCA-R2 C10600-10B digital charged-couple device (CCD) camera (Hamamastu, Japan) at 3 Hz. The average fluorescence of all pixels for each time point in a defined region of interest (ROI) was subtracted from the average background fluorescence of the same size ROI within the brain region. The resulting fluorescence value for each time point was defined as Ft. % $\Delta F/F_0 = (F_t - F_0)/F_0 \times 100$, where $F_0$ corresponds to average of 10 frames of background-subtracted baseline fluorescence before ATP application. All images were processed and quantified using Zen and Fiji (Image J).

**Arc light imaging**. Imaging of freshly dissected brain explants was performed on a Zeiss Axio Examiner upright microscope using a W Plan Apochromat ×40 N.A. 1.0 water immersion objective (Zeiss, Germany). ArcLight was excited with a 470 nm LED (Zeiss). The objective C-mount image was projected onto the 80 × 80 pixel chip of a NeuroCCD-SM camera controlled by NeuroPlex software (Red-ShirtImaging, Decatur, GA). Images were recorded at a frame rate of 100 Hz, and depicted optical traces were spatial averages of intensity of all pixels within the ROI, with signals processed as previously reported[47]. Statistical analysis and plotting of the data were performed using R. Sleep-deprived flies and sleep-replete controls were dissected within a few minutes of sleep-deprivation and recorded at the same ZT.

**Data availability**. We declare that all data supporting the findings of this study are available within the article and its Supplementary Information files or from the corresponding author upon reasonable request.

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

## Acknowledgements

We thank the Tsinghua Fly Center and Bloomington Stock Center for fly stocks; Gerry Rubin and JaneliaFlyLight team for fly stocks and expression data of *LexA* lines (e.g., Supplementary Fig. 6); Chuan Zhou for *R41A01* expression data; Alison Howard for administrative support; and members of the Janelia fly facility for technical support. We also thank Wei Xie and Junhai Han for technical support and discussion. This work was supported by the Howard Hughes Medical Institute and the Janelia Visitor Program, and College of Arts and Sciences, University of San Diego (to D.S.). Work in the laboratory of Y.P. was also supported by the Fundamental Research Funds for the Central Universities (3231004206), the National Natural Science Foundation of China (31571093, 31622028), the Natural Science Foundation from Jiangsu Province (BK20150597, BK20160025), and the Thousand Young Talents Program in China. Work in the laboratory of M.N.N. was supported in part by the National Institute of Neurological Disorder and Stroke and National Institute of General Medicine, NIH

(R01NS055035, R01NS056443, R01NS091070, and R01GM098931), and The Kavli Institute for Neuroscience at Yale.

## Author contributions

D.S., B.S.B., M.N.N., and Y.P. conceived the experiments, interpreted the results, and wrote the manuscript. D.S., D.C., and Y.P. carried out most of the experiments and analyzed data. X.J. carried out the Arclight experiment; N.C. and C.H. helped with sleep experiments; J.C. and M.S. helped with GCamp imaging.

## Additional information

**Competing interests:** The authors declare no competing financial interests.

