## [Peer Review File · Nature Communications]

Reviewers' Comments:

Reviewer #1 (Remarks to the Author):

These authors examine differences in courtship behavior between sleep-deprived male and female flies and differences in sleep behavior between male and female flies. They also implicate the well-studied male P1 neurons (that express both fruM and dsxM) in modulating sleep, adding another behavior to the repertoire of behaviors now under regulation of these important central brain neurons. And, they determine the neurons and genes downstream of P1 that affect sleep. This study therefore adds to the growing literature on sexual dimorphisms and behavior and will be important not only for those working on *Drosophila*, but more broadly for neuroscientists studying sexual differences in behavioral states and drive. Nonetheless there are major problems with this manuscript at present that must be addressed prior to publication:

1) The paper claims to be about action selection – choosing one behavioral action over another – but the authors don't study, for example, how locomotor circuits are recruited or activated differently during courtship versus sleeping, nor do they really examine how males choose between sleeping versus courting (for example, a choice between sleep and courtship in sleep-deprived males). Instead the authors' major findings are related to how male-specific P1 neurons control sleep in *Drosophila*. This is in and of itself interesting given the known role of P1 in song production and courtship – the authors therefore don't need to make the paper about action selection but should instead focus the Intro on motivating their actual results.

2) The authors first find that males sleep deprived for 16 hours reduce courtship behavior but females sleep deprived for a similar amount of time do not change their receptivity. I don't understand this comparison – on the one hand the authors are measuring a behavior in males that requires continual motion (chasing) and wing vibration. In females, they simply measure time to mating. These are two completely different metrics and it is no surprise that one (which is energetically demanding) is affected by sleep loss (male courtship) while the other is not. I can't understand why this result suggests that "sleep deprivation suppresses sexual behavior in male flies but not in female flies". That seems to be the wrong conclusion to come to from this set of experiments.

3) The authors find that males sleep more than females, on average (which as stated above may reflect the different energetic needs of males versus females) and that this difference goes away by manipulating the sex determination genes (*tra/dsx/fru*). This result does not seem all that surprising given the observed difference in average sleep between males and females. The authors continually discuss the sexual dimorphism but never address the mechanisms underlying it. Which specific neurons cause males to sleep more than females? While the authors find that P1 and DN1 (male-specific neurons) affect sleep in males, they do not go one step further to connect the results from P1 and DN1 to the differences in male and female sleep.

4) The authors use the R71-LexA/dsx-Gal4 intersection which drives expression in a subset of the male-specific P1 neurons. When thermogenetically activated, this causes males to sleep less compared with controls, with phenotypes that look like sleep fragmentation. Expressing a neural silencer in these same neurons causes males to sleep more. The authors conclude that “These results indicate that increased sexual arousal suppresses sleep in male flies, while decreased sexual arousal promotes sleep in males.” No - the result only indicates that the neurons labeled in R71-LexA/dsx-Gal4 have a role in sleep – the results as presented have nothing to do with the role of sexual arousal in sleep. The same criticism applies to their interpretation of the pC1 activation/silencing experiments in females.

5) The DN1 neurons were already implicated in regulating circadian rhythms in male courtship behavior (Fujii and Amrein 2010). This paper adds to that result by showing that DN1 neurons are downstream of P1 neurons in regulating sleep (and that DN1 neurons have to express fruM, which in turn regulates Dh31, a nice result). In contrast, sleep deprivation reduces baseline calcium levels (measured via GCaMP) in P1 neurons (the authors should show the DF/F responses as a function of hours of sleep deprivation for this result to be convincing).

Several questions remain – what levels of activity in P1 are needed to promote sleep versus courtship and is it all or some P1 neurons that participate in these different behaviors? By looking more carefully at behavior the authors might be able to conclude more on this front. For example, the authors could measure courtship-related behaviors in single flies (wing extension; see Hoopfer et al. eLife) while also tracking them in sleep/activity monitors – this would allow them to measure the dynamics of courtship activity caused by P1 activation relative to sleep activity. If P1 drives DN1 to drive waking, then does this circuit feedback to P1 (since P1 activity is low when males are sleepy)? Why didn't the authors put P2X2 in DN1 and examine P1 responses? This would have given the authors more mechanism underlying the interaction between sleep drive and courtship drive.

6) A recent paper (Zhang...Crickmore Neuron) examined male courtship drive and dopamine neurons – the authors don't reference this paper, but doing so would allow them to discuss how behavioral state modulates P1 (which was carefully examined in the Crickmore paper) which then in turn affects courtship activity and now (this paper) a new feedback mechanism (modulating behavioral state itself via DN1).

6) This is a more minor criticism, but overall the paper reads like a jumble of results rather than a carefully thought out story. The results in Figure 6, for example, are nice and seem to be the result of a lot of work, but they don't really fit into the story as is – the reader gets the feeling the authors stuck together all the data that was collected whether it fit or not. The paper could be improved by more thoughtful rewriting.

Reviewer #2 (Remarks to the Author):

This manuscript by Sitaraman et al. investigates the interaction between sex and sleep drive in male and female flies. The authors show that severely sleep-deprived male flies display reduced courtship towards females (but females are equally receptive even after sleep deprivation). Next, they show that artificially

activating P1 neurons in males (directly promoting male courtship) acutely perturbs sleep (while a similar experiment, activating female-receptivity neurons has no effect on sleep). The effect of P1 activation could be blocked by concurrent silencing of DN1 wake-promoting circadian neurons, and indeed DN1s could be activated *ex-vivo* by artificial stimulation of P1s, indicating a potential functional connection between the two. Next, the authors investigate the neurotransmitter that P1s use to promote wakefulness (Ach). In the last part of the manuscript, the authors embark on a study aimed at discovering the basis of sexually dimorphic sleep patterns (it is known that males sleep more than female flies), and which combinations of sex-determining factors acts in which cells to produce this difference –the results suggests that this effect is mediated by MB and DN1s.

Albeit this paper contains high-quality experiments (all immunostainings are beautiful and the genetic manipulations are of high quality and well-controlled) I found this manuscript very difficult to follow and convoluted, the conclusions are generally over-stated and not well supported. The main problem is that the story does not come together in a coherent, unitary narrative; rather, the authors attempt to combine two largely diverging lines of research (one about P1 neurons and their ability to promote wakefulness in males and one about the genetic control of sexually dimorphic sleep patterns). Each part of the work has some convincing data, but these two lines do not come together in a coherent narrative, on the other hand each side of the story is not sufficiently compelling to recommend publication by itself.

I limited my detailed analysis to the first of the two “conjoined” stories (Figures 1-5, Suppl. Figures 1-6). This first part represents the bulk of the paper and is potentially interesting, but has a number of critical flaws that make the conclusions unconvincing. While a strong case can be made that artificial activation of P1 neurons can override sleep drive probably acting on wake-promoting DN1s, the conclusion that P1s are required for normal sleep and that sleep drive affects excitability of P1s are tentative at best. P1 neurons are strongly activated in response to olfactory and gustatory stimuli from females (see Clooney et al 2015 for example), and, when activated, can promote robust courtship and even fighting (see Hoopfer et al.). It is not surprising then that strong artificial activation would disturb the flies’ sleep. Conversely, all sleep and imaging studies reported here are conducted on isolated animals, i.e. in the absence of stimuli that activate P1s; it is hard to imagine what the normal function of P1s in sleep may be in these conditions, and it is hard to compare the activity of P1s in sleep deprived vs non-sleep deprived animals in the absence of normal stimuli. I see little evidence for the main claim here, that “These studies reveal that... reciprocal inhibition between sex and sleep circuitries is responsible for action selection of these behaviors”. The data is also at odds with published work on a few key points

Major points:

1. Figure 2. Here the authors show that artificial activation of P1s, known to cause strong induction of courtship behavior, perhaps not surprisingly disturb flies’ sleep (at least for a couple of days). The data on the effects on sleep of P1 activation is convincing (at least in the first day of heat –Figure 2c), but the data on P1 inactivation by TNT is not equally strong, in particular when various genotypes seem to have quite different levels of baseline sleep (ranging from 700 to more than 1000 minutes for males, see Figure 6). This result is not intuitive, as –to my knowledge- P1s have never before directly linked to sleep. To make this point more convincing, the authors should use shibire-TS to acutely inactivate P1 neurons (as they do in other experiments) and measure the effects on sleep, similar to what they do in Figure 1c for activation by TRPA1. If the result still holds, a careful discussion of its significance is going to be

important: P1s are supposed to regulate male courtship in response to female-derived sensory stimuli, while sleep experiments are performed on isolated males.

2. Figure 2. Do wild type flies sleep the same amount at 21.5 as 28.5°C? According to published work (see for example Parisky et al, 2016), high temperature (29°C) changes considerably sleep patterns in fruit flies. Are these data compatible with the published work? If not, this should be commented upon.

3. Figure 3. The observation that silencing all these key neuronal cell types has no effect on sleep is surprising. In fact, this seems to be in direct contrast with published data. For example Guo et al, 2016 show that silencing DN1 (using 18H11, the same driver used here!) reduced sleep across all ZTs, while a different paper (Liu et al, 2013) showed that silencing the fan shape body also reduced sleep. Who is correct here?

4. Figure 4. This experiment has a number of shortcomings, lacks appropriate controls, and it is difficult to interpret –yet it is used as a pivot to argue that sleep drive reduces courtship by directly affecting P1 excitability. How is this experiment done? Are these ex-vivo preparations or intact flies? What is the significance of the “episodes” observed? Recordings on P1s during male-female encounters show large stimulus-evoked activity and very little baseline “bumps” like the ones shown here. It is difficult to compare activity in sleep deprived and non-deprived P1s in the absence of stimulation, this experiment should be done in an intact animal interacting with a female for example as shown in Clowney et al, 2015. It is also important to show an appropriate control, i.e. a neuron the response of which is not modulated by sleep deprivation (to account for potential nonspecific effects –like change in pH, cytotoxicity or what not- that may be brought about by this very prolonged sleep deprivation). Also, the argument that P1-TRPA1 can overcome sleep drive is not convincing. TRPA1 mediated activation of PN1s produces very robust courtship; it is not surprising that it can overcome SD. At the very least the authors could establish a dose-response of P1 activation using for example optogenetics, and show that higher thresholds are required to activate P1s after SD. I would also note that here the authors argue that DN1s are downstream of P1s, yet acute silencing of DN1s cannot suppress TRPA1-induced courtship.

5. Page 9 line 14. “These data support the hypothesis that DN1 circadian clock neurons participate in a shared circuit for controlling sex and sleep” I see no evidence at this point in the paper that DN1s participate in anything other than sleep/arousal. In fact, silencing DN1s has no effect on courtship as quantified in this work (Supplementary Figure 6). The authors would be advised to look carefully into courtship rhythms, which have been shown to depend on DN1s (see Fujii and Amrein, 2010).

Minor points:

6. Figure 1. The authors report that sleep deprivation (SD) for 8 hours or 12 hours between ZT12-ZT0 had no effect on sleep, while 12 hours SD between ZT16-ZT4 or 16 hours SD had an effect on male courtship. They also report that 16 hrs SD, but not 8 hrs SD produced sleep rebound. Do all conditions that induce courtship suppression also induce sleep rebound? This may be important to understand why some sleep deprivation protocols, but not others, are effective in reducing courtship drive.

7. Figure 1. The null hypothesis here has to be that the two behaviors do not directly interact (courtship and sleep). Even if this is the case, flies will still not perform courtship behavior while sleeping (i.e. the two are mutually exclusive). The courtship assay immediately follows sleep deprivation, but from the data it would be hard to establish if during this assay the flies are awake. It would be important to show that, while flies court much less, they do indeed move around the chamber.

8. Page 7 line 23. The statement that “The results suggest that sexual arousal positively regulates female sleep” is contradictory. The effect on arousal of pC1 and pCd can only be an inference; a more direct statement of the observed result should be used instead.

9. Page 10 line 11. “The above results identify a neural substrate for sexual arousal to regulate sleep.” I think it is more appropriate to say that forced activation of P1 can overcome sleep drive by activating wake-promoting DN1s. At this point no evidence has been shown that sleep drive has an effect on P1s or DN1s (the authors report that acute silencing of DN1s has no effect on sleep and do not show acute inactivation of P1s see above)

10. Page 12 line 16. “We also found that females pan-neuronally expressing microRNAs targeting fruM (c155>UAS-fruMi) sleep as much as males of the same genotype, while control females expressing a scrambled version (UAS-fruMiScr) sleep less than males (Fig. 6b and 6c).” Has this tool been published before? If so the publication should be cited, otherwise essential controls on the efficacy of this reagent should be included.

Reviewer #3 (Remarks to the Author):

The following study addresses interplay between sleep and mating behavior at the level of single genes and circuits in fruit flies. Study offers interesting findings that will be of interest for the wide audience. However before recommending this work for publication several questions/comments need to be addressed by authors.

1. Authors like to frame their findings in terms of action selection for sleep and courtship behavior yet sleep is not classically defined as action but rather as state.
2. In Fig 1a authors make a point that not only duration of the sleep but also timing of the sleep is important for vigor of the male courtship. Yet in Fig1a there is no statistical comparison between ZT0 and ZT4 for the same 12 hr sleep deprivation.
3. In Fig1b-e it is not clear how authors take the baseline. How do they exactly choose the time period for baseline?
4. For supplementary Fig1e. authors show that female flies do not have sleep rebound. Is it a known fact? If not authors should state that it is their discovery, otherwise provide reference.

5. In Fig2a. It would be nice to show high-resolution cellular imaging of these neurons.
6. On page 7, lines 7-9 authors state that “Furthermore, sleep reduction in P1-activated male flies is not due to changes in general locomotor activity, as the experimental and control genotypes are equally active while awake (Supplementary Fig. 3).” Yet locomotor activity is measured with the same tools that measures sleep (beam crossing), however it would be better to evaluate movement changes in these flies using alternative method like simple video tracking and getting the speed and distance data from individual flies. This way authors can monitor small changes in movement of animals that may have changed in experimental group.
7. In Fig 2.b-d control without LexAop2-FlpL is missing.
8. Authors assume that activating P1 neurons in males for 36hr period (Fig2b,c) reduces sleep duration. However such a prolonged activation of neurons might induce plasticity and reduce P1 neuron impact on its synaptic partners. Therefore if authors want to claim that activation of P1 neuron increases sexual drive they need to also check male courtship behavior at the beginning and at the end of 36hr period.
9. Based on Fig4b,c authors claim that sleep deprivation reduces activity of P1 neurons. However they plot standard error and power spectrum of the imaged signal. It was not clear to me why authors choose to use these measures. More direct way of measuring the change in these flies would be to plot mean of $\Delta F/F$ with standard error in control and experimental flies.
10. Based on results of knockdown of dVAcHT gene expression authors conclude that this neurotransmitter is mediating P1 activity on its synaptic partners, yet they never showed that the gene knockdown indeed took place. Authors should use immunohistochemistry or RNA labeling tools to make sure that the gene was really expressed at lower levels than in controls.
11. In Fig 5a most of the gene knockdown may reach statistical significance compared to the controls. Could authors maybe comment on this? Could this be a nonspecific effects of the RNAi knockdown?
12. Authors nicely demonstrate that fruM is necessary for sleep regulation in DN1 neurons. It would be desirable to see the sufficiency experiments too. Authors could express fruM in female DN1 neurons and see if they can convert female sleep into male.

Our responses include a summary part (below in *italics*) and a point-to-point part immediately after the comments (in blue).

Since the original submission of our manuscript we have made substantial additional changes to the manuscript in terms of new experimental data and writing. Because revisions are extensive, as outlined below, changes to the text have not been individually highlighted in the main text. We have indicated the line numbers and made reference to additional figures wherever appropriate.

There are two major types of new experiments and a few minor ones.

Major new experiments:

- 1. We have performed detailed video tracking analysis of P1 activated males and analyze courtship and sleep related parameters, using different temperatures (25.5°C, 27°C, 28.5°C and 30°C) to activate P1 neurons and compare their effects on sleep and courtship. Interestingly we find that P1 activation at low levels (27°C) is sufficient to induce wake effects but a higher level of activation (28.5°C) is required to induce courtship behaviors. We have conducted these experiments using two distinct methods of P1 targeting to ensure that these effects are independent of genetic strategy. These experiments clearly indicate that P1 neurons regulate male courtship and sleep in a threshold-dependent manner. These set of experiments have now been added as **new Fig. 3.***
- 2. We tested P1's Calcium response to DNI activation by P2X2, and we found that P1 neurons do respond to DNI activation, and P1-DNI neurons form a positive feedback loop supported by mutually excitatory interactions. These interactions likely form a self-reinforcing loop that allows persistence of arousal to allow the animal to stay awake while performing important social behaviors like courtship. These results have provided new mechanistic insight into the interactions between sleep-drive and courtship, and have now been added as **new Fig. 6.***

*Minor ones: we have performed almost all additional experiments as requested by reviewers, e.g., video tracking of locomotion, and RNAi efficiency tests. These new experiments were incorporated into the manuscript appropriately as **Fig. 2g, Fig. S2, Fig. S4, Fig. S8c, Fig. S9, Fig. S10 and Movie S4.***

We also made a working model (Fig. 10) to summarize our major findings.

Reviewer #1 (Remarks to the Author):

These authors examine differences in courtship behavior between sleep-deprived male and female flies and differences in sleep behavior between male and female flies. They also implicate the well-studied male P1 neurons (that express both fruM and dsxM) in modulating sleep, adding another behavior to the repertoire of behaviors now under regulation of these important central brain neurons. And, they determine the neurons and genes downstream of P1 that affect sleep. This study therefore adds

to the growing literature on sexual dimorphisms and behavior and will be important not only for those working on *Drosophila*, but more broadly for neuroscientists studying sexual differences in behavioral states and drive. Nonetheless there are major problems with this manuscript at present that must be addressed prior to publication:

1) The paper claims to be about action selection – choosing one behavioral action over another – but the authors don't study, for example, how locomotor circuits are recruited or activated differently during courtship versus sleeping, nor do they really examine how males choose between sleeping versus courting (for example, a choice between sleep and courtship in sleep-deprived males). Instead the authors' major findings are related to how male-specific P1 neurons control sleep in *Drosophila*. This is in and of itself interesting given the known role of P1 in song production and courtship – the authors therefore don't need to make the paper about action selection but should instead focus the Intro on motivating their actual results.

We are gratified by the reviewer's acknowledgement of the significance of our findings regarding the role of P1 neurons in sleep. We have made requisite changes in our introduction section (e.g., Line: 86-92) and title of the paper to accurately reflect the importance of these results.

2) The authors first find that males sleep deprived for 16 hours reduce courtship behavior but females sleep deprived for a similar amount of time do not change their receptivity. I don't understand this comparison – on the one hand the authors are measuring a behavior in males that requires continual motion (chasing) and wing vibration. In females, they simply measure time to mating. These are two completely different metrics and it is no surprise that one (which is energetically demanding) is affected by sleep loss (male courtship) while the other is not. I can't understand why this result suggests that “sleep deprivation suppresses sexual behavior in male flies but not in female flies”. That seems to be the wrong conclusion to come to from this set of experiments.

We have made changes in the text to accurately reflect the difference in measurements between female and male courtship index (e.g., Line 127-129). Even though we understand the reviewer's concern regarding the difference between male and female sexual behaviors, our conclusion that “sleep deprivation suppresses sexual behavior in male flies but not in female flies” is strongly supported by established behavioral experiments. Female courtship behaviors have been traditionally measured and defined by the time to copulation as used in this study and less demanding as compared to males.

3) The authors find that males sleep more than females, on average (which as stated above may reflect the different energetic needs of males versus females) and that this difference goes away by manipulating the sex determination genes (*tra/dsx/fru*). This result does not seem all that surprising given the observed difference in average sleep between males and females. The authors continually discuss the sexual dimorphism but never address the mechanisms underlying it. Which specific neurons cause males to sleep more than females? While the authors find that P1 and DN1 (male-specific neurons) affect sleep in males, they do not go one step further to connect the results from P1 and DN1 to the differences in male and female sleep.

We think the reviewer may have mis-interpreted some of our data. In our manuscript, we show that FruM functions within MB (mushroom body) and DN1 neurons, but not P1 neurons, to promote sleep (Figure 9). We also show that FruM likely regulates wake-inducing neuropeptide Dh31 in DN1 neurons to confer its function on sleep (e.g., Line 379- 389).

In short FruM functions within distinct neural substrates to regulate courtship and sleep. The role of FruM in DN1 and MB in regulating sleep is novel as reported in our study. We have made a stronger emphasis on this finding in our revision (e.g., line 365-378 and 468-472)

4) The authors use the R71-LexA/dsx-Gal4 intersection which drives expression in a subset of the male-specific P1 neurons. When thermogenetically activated, this causes males to sleep less compared with controls, with phenotypes that look like sleep fragmentation. Expressing a neural silencer in these same neurons causes males to sleep more. The authors conclude that “These results indicate that increased sexual arousal suppresses sleep in male flies, while decreased sexual arousal promotes sleep in males.” No - the result only indicates that the neurons labeled in R71-LexA/dsx-Gal4 have a role in sleep – the results as presented have nothing to do with the role of sexual arousal in sleep. The same criticism applies to their interpretation of the pC1 activation/silencing experiments in females.

Since, activation of the P1 neurons with dTRPA1 targeted by our intersectional genetics using lexA-Gal4-flp system and split-Gal4 system have been previously implicated in sexual arousal we made the conclusion that sexually aroused flies (as a result of P1 activation) alter sleep. We understand that the reviewers might see this as an extended interpretation so we have re-written the interpretations to indicate the link between P1 activation and sleep effects without referring to this as a sexually aroused state. We also present data showing that inhibition of these neurons with TNT (Figure 2d) and Shibire ts1 (Figure 2g) associated with diminished courtship has opposite effect on sleep.

5) The DN1 neurons were already implicated in regulating circadian rhythms in male courtship behavior (Fujii and Amrein 2010). This paper adds to that result by showing that DN1 neurons are downstream of P1 neurons in regulating sleep (and that DN1 neurons have to express fruM, which in turn regulates Dh31, a nice result). In contrast, sleep deprivation reduces baseline calcium levels (measured via GCaMP) in P1 neurons (the authors should show the DF/F responses as a function of hours of sleep deprivation for this result to be convincing). Several questions remain – what levels of activity in P1 are needed to promote sleep versus courtship and is it all or some P1 neurons that participate in these different behaviors? By looking more carefully at behavior the authors might be able to conclude more on this front. For example, the authors could measure courtship-related behaviors in single flies (wing extension; see Hoopfer et al. eLife) while also tracking them in sleep/activity monitors – this would allow them to measure the dynamics of courtship activity caused by P1 activation relative to sleep activity. If P1 drives DN1 to drive waking, then does this circuit feedback to P1 (since P1 activity is low when males are sleepy)? Why didn't the authors put P2X2 in DN1 and examine P1 responses? This would have given the authors more mechanism underlying the interaction between sleep drive and courtship drive.

We really thank the reviewer for these observations. We have addressed each of the concerns as follows:

(1) Regarding our data of P1 activity in SD males, we actually used the genetically encoded voltage sensor ArcLight that has been validated for functional optical imaging of neuronal activity simultaneously in groups of genetically targeted neurons instead of the sensor for Calcium (GCaMP); ArcLight has much faster kinetics than GCaMP6, and is ideal for detecting spontaneous neural activity as validated in our recent paper describing sleep and arousal microcircuits and clock neurons (Cao et al., 2013, Sitaraman et al., 2015). As suggested we performed additional analysis of the $\Delta F/F$ responses (Figure S9), which further confirms our conclusion that sleep deprivation inhibits P1 activity.

(2) We have now performed detailed video tracking analysis of P1 activated males and analyze courtship and sleep related parameters, using different temperatures (25.5°C, 27°C, 28.5°C and 30°C) to activate P1 neurons and compare their effects on sleep and courtship. These set of experiments have now been added as Figure 3. Interestingly we find that P1 activation at low levels (27°C) is sufficient to induce wake effects but a higher level of activation (28.5°C) is required to induce courtship behaviors. We have conducted these experiments using two distinct methods of P1 targeting to ensure that these effects are independent of genetic strategy.

(3) We appreciate the suggested experiment testing if DN1 neurons feedback to P1 neurons. We have now performed these experiments and interestingly we find that P1-DN1 neurons form a positive feedback loop supported by mutually excitatory interactions. These interactions likely form a self-reinforcing loop that allows persistence of arousal to allow the animal to stay awake while performing important social behaviors like courtship. These results have provided new mechanistic insight into the interactions between sleep-drive and courtship (Figure 6).

In addition to presenting these new findings we have also added details in the discussion related to the sensory inputs into P1 neurons and sleep regulating outputs from DN1 neurons (e.g., line 410-415 and 478-482). P1 neurons have been studied in great detail and considered to be critical for integration of courtship relevant sensory signals, hence it's highly likely that in addition to DN1 input, sleep affects P1 neurons by gating these sensory responses.

6) A recent paper (Zhang...Crickmore Neuron) examined male courtship drive and dopamine neurons – the authors don't reference this paper, but doing so would allow them to discuss how behavioral state modulates P1 (which was carefully examined in the Crickmore paper) which then in turn affects courtship activity and now (this paper) a new feedback mechanism (modulating behavioral state itself via DN1).

We thank the reviewers for this observation and have now carefully examined this paper and integrated its findings to our paper in light of DN1 input into P1 neurons (e.g., line 410-415).

6) This is a more minor criticism, but overall the paper reads like a jumble of results rather than a carefully thought out story. The results in Figure 6, for example, are nice and seem to be the result of a lot of work, but they don't really fit into the story as is –

the reader gets the feeling the authors stuck together all the data that was collected whether it fit or not. The paper could be improved by more thoughtful rewriting.

We have now made substantial changes to results and discussion section to more accurately reflect the motivation behind the experiments and significance of the results. We strongly believe that these experiments elaborate the neural and genetic basis of sexually dimorphic behavioral choice between sleep and sex.

Reviewer #2 (Remarks to the Author):

This manuscript by Sitaraman et al. investigates the interaction between sex and sleep drive in male and female flies. The authors show that severely sleep-deprived male flies display reduced courtship towards females (but females are equally receptive even after sleep deprivation). Next, they show that artificially activating P1 neurons in males (directly promoting male courtship) acutely perturbs sleep (while a similar experiment, activating female-receptivity neurons has no effect on sleep). The effect of P1 activation could be blocked by concurrent silencing of DN1 wake-promoting circadian neurons, and indeed DN1s could be activated ex-vivo by artificial stimulation of P1s, indicating a potential functional connection between the two. Next, the authors investigate the neurotransmitter that P1s use to promote wakefulness (Ach). In the last part of the manuscript, the authors embark on a study aimed at discovering the basis of sexually dimorphic sleep patterns (it is known that males sleep more than female flies), and which combinations of sex-determining factors acts in which cells to produce this difference –the results suggests that this effect is mediated by MB and DN1s.

Albeit this paper contains high-quality experiments (all immunostainings are beautiful and the genetic manipulations are of high quality and well-controlled) I found this manuscript very difficult to follow and convoluted, the conclusions are generally over-stated and not well supported. The main problem is that the story does not come together in a coherent, unitary narrative; rather, the authors attempt to combine two largely diverging lines of research (one about P1 neurons and their ability to promote wakefulness in males and one about the genetic control of sexually dimorphic sleep patterns). Each part of the work has some convincing data, but these two lines do not come together in a coherent narrative, on the other hand each side of the story is not sufficiently compelling to recommend publication by itself.

We would like to thank the reviewer for the observations related to our high-quality immunostaining experiments and well controlled behavioral assays. Sex and sleep are sexually dimorphic behaviors and hence the selection mechanisms are sex specific as well. Hence, the focus of this study is not only sexually dimorphic decision-making and behavioral selection but also the genes that make these circuits distinct. We understand that this was not explicitly clear in our earlier submission so we have rewritten the introduction and conclusion extensively to make these parts more cohesive.

I limited my detailed analysis to the first of the two “conjoined” stories (Figures 1-5, Suppl. Figures 1-6). This first part represents the bulk of the paper and is potentially interesting, but has a number of critical flaws that make the conclusions unconvincing. While a strong case can be made that artificial activation of P1 neurons can override sleep drive probably acting on wake-promoting DN1s, the conclusion that P1s are required for normal sleep and that sleep drive affects excitability of P1s are tentative at best. P1 neurons are strongly activated in response to olfactory and gustatory stimuli from females (see Clooney et al 2015 for example), and, when activated, can promote robust courtship and even fighting (see Hoopfer et al.). It is not surprising then that strong artificial activation would disturb the flies’ sleep. Conversely, all sleep and imaging studies reported here are conducted on isolated animals, i.e. in the absence of stimuli that activate P1s; it is hard to imagine what the normal function of P1s in sleep may be in these conditions, and it is hard to compare the activity of P1s in sleep deprived vs non-sleep deprived animals in the absence of normal stimuli. I see little evidence for the main claim here, that “These studies reveal that... reciprocal inhibition between sex and sleep circuitries is responsible for action selection of these behaviors”. The data is also at odds with published work on a few key points

We would like to reiterate that our conclusion is based on the following experimental evidence provided in the manuscript:

- 1) Sleep-deprived flies that are sleepy (have a higher sleep-drive) court less and have diminished P1 neural activity (Figure 1 and 7a, b, c)
- 2) Sleep-deprivation induced courtship deficit can be rescued by P1 activation (Figure 7d)
- 3) Flies with artificial activation of P1 neurons have high courtship drive but reduced sleep-drive (figure 2 and 3)
- 4) P1 neurons induce wake activity by interacting with DN1 neurons (Figure 5)
- 5) P1 and DN1 neurons have reciprocal excitatory interactions (supported by new figure 6)
- 6) Inhibition of P1 neurons using TNT and Shi^{ts1} promotes sleep suggesting the necessity of these neurons in sleep-regulation (Figure 2).

We strongly believe that our behavioral data and circuit analysis support our conclusion but we have now edited the use of the term reciprocal inhibition as it can be misinterpreted. Our new data showing that P1 and DN1 neurons form mutually excitatory synaptic connections provides critical mechanistic insight into the “reciprocal control” of sleep and courtship (Figure 10).

Major points:

1. Figure 2. Here the authors show that artificial activation of P1s, known to cause strong induction of courtship behavior, perhaps not surprisingly disturb flies’ sleep (at least for a couple of days). The data on the effects on sleep of P1 activation is convincing (at least in the first day of heat –Figure 2c), but the data on P1 inactivation by TNT is not equally strong, in particular when various genotypes seem to have quite different levels of baseline sleep (ranging from 700 to more than 1000 minutes for males, see Figure 6). This result is not intuitive, as –to my knowledge- P1s have never before directly linked to sleep. To make this point more convincing, the authors should use shibire-TS to acutely inactivate P1 neurons (as they do in other

experiments) and measure the effects on sleep, similar to what they do in Figure 1c for activation by TRPA1. If the result still holds, a careful discussion of its significance is going to be important: P1s are supposed to regulate male courtship in response to female-derived sensory stimuli, while sleep experiments are performed on isolated males.

We understand that reviewers concern regarding TNT and have repeated the inhibition experiments with shibire-TS1 (Figure 2G). We find that conditional inhibition and activation of P1 neurons have opposite effects on sleep. Since, *Shibire ts1* was induced by temperature shift in adult flies these data also rule out any developmental effects of P1 inhibition.

2. Figure 2. Do wild type flies sleep the same amount at 21.5 as 28.5°C? According to published work (see for example Parisky et al, 2016), high temperature (29°C) changes considerably sleep patterns in fruit flies. Are these data compatible with the published work? If not, this should be commented upon.

We have presented a lot of sleep data at lower or higher temperatures in this manuscript as the most reliable effectors to activate and inhibit neurons rely on temperature shifts. In general, temperature affects sleep pattern in a lot of genotypes including the wild type, but this effect is somewhat variable and highly background specific. Thus, for each sleep experiments we did, we have used control lines with the most similar genetic background. Reviewers have themselves acknowledged our use of extensive controls to obtain reliable data and we feel that these controls accurately reflect temperature effects. All controls were chosen to eliminate any potential confounding genetic background effects, as well as the number and chromosomal insertion sites of mini-white-containing transgenes. We have now added more detailed description of these control genotypes to elaborate this point.

3. Figure 3. The observation that silencing all these key neuronal cell types has no effect on sleep is surprising. In fact, this seems to be in direct contrast with published data. For example Guo et al, 2016 show that silencing DN1 (using 18H11, the same driver used here!) reduced sleep across all ZTs, while a different paper (Liu et al, 2013) showed that silencing the fan shape body also reduced sleep. Who is correct here?

We have now made a change in the manuscript to explain our inability to observe sleep phenotypes when some of the sleep-regulating neurons are inhibited (e.g., line 245-247). We believe that some of these sleep-regulating neurons have relatively reduced basal neural activity and further inhibition in sleep-replete animals fails to produce a sleep-phenotype. Activating of P1 neurons activates the DN1 neurons making the silencing effect stronger and more effective.

The LexA/LexAop system is weaker than the GAL4/UAS system and we have used milder temperatures for shibire so we suspect that we are getting partial silencing. These factors might also have contributed to the lack of phenotypes.

In comparison, Guo et al used TNT (not inducible) and Liu et al show that manipulating dopamine receptor in dFB neurons reduced sleep. We have not been able to find evidence of use of shibire *ts1* in these neurons driven by LexA/LexAop system.

4. Figure 4. This experiment has a number of shortcomings, lacks appropriate controls, and it is difficult to interpret –yet it is used as a pivot to argue that sleep drive reduces courtship by directly affecting P1 excitability. How is this experiment done? Are these ex-vivo preparations or intact flies? What is the significance of the “episodes” observed? Recordings on P1s during male-female encounters show large stimulus-evoked activity and very little baseline “bumps” like the ones shown here. It is difficult to compare activity in sleep deprived and non-deprived P1s in the absence of stimulation, this experiment should be done in an intact animal interacting with a female for example as shown in Clowney et al, 2015. It is also important to show an appropriate control, i.e. a neuron the response of which is not modulated by sleep deprivation (to account for potential nonspecific effects –like change in pH, cytotoxicity or what not- that may be brought about by this very prolonged sleep deprivation). Also, the argument that P1-TRPA1 can overcome sleep drive is not convincing. TRPA1 mediated activation of P1s produces very robust courtship; it is not surprising that it can overcome SD. At the very least the authors could establish a dose-response of P1 activation using for example optogenetics, and show that higher thresholds are required to activate P1s after SD. I would also note that here the authors argue that DN1s are downstream of P1s, yet acute silencing of DN1s cannot suppress TRPA1-induced courtship.

We thank the reviewer for these observations. We have addressed these multiple concerns as follows:

(1) We used ArcLight, an optical sensor of membrane potential developed and well characterized by the Nitabach lab, to record neural activity in intact fly brain and explant preparation. We have now added details of the ArcLight recordings of P1 neurons in the methods sections. The changes in fluorescence intensity is indicative of voltage changes and we have recently validated the use of ArcLight in recording spontaneous neural activity of sleep-regulating neurons of the mushroom body (Sitaraman et al 2015). We did this experiment in sleep-deprived males without presenting a female to monitor the spontaneous activity of P1 neurons. We think repeating such experiment while presenting a female would not add much information, as it has been well established that P1 activity is correlated with courtship behavior to female, and we do show that sleep-deprived males have reduced courtship to females.

(2) The reviewer asks for a control i.e. a neuron whose activity is not modulated by SD. We have published data on subsets of MB neurons (γ main Kenyon cells), which have no response to SD and are indistinguishable from non-sleep deprived controls. Further we have also identified sleep promoting MB neurons which show increased activity (γ 2 α '1 MB output neurons and γ d Kenyon cells) after SD (Sitaraman et al 2015). Thus, the decrement of activity in P1 neurons is not a non-specific effect and specifically altered by sleep deprivation.

(3) The reviewer asked to test the outcome of a dose-response of P1 activation on sleep and courtship. We have now performed these experiments by using dTrpA1 in P1 neurons at different temperatures (25.5°C, 27°C, 28.5°C and 30°C), and find that P1 activation at low levels (27°C) is sufficient to induce wake effects but a higher level of activation (28.5°C) is required to induce courtship behaviors (New Figure 3). We have conducted these experiments using two distinct methods of P1 targeting to

ensure that these effects are independent of genetic strategy. We thank the reviewer for this suggestion as it provides novel insight into P1's role in arousal and courtship.

5. Page 9 line 14. "These data support the hypothesis that DN1 circadian clock neurons participate in a shared circuit for controlling sex and sleep" I see no evidence at this point in the paper that DN1s participate in anything other than sleep/arousal. In fact, silencing DN1s has no effect on courtship as quantified in this work (Supplementary Figure 6). The authors would be advised to look carefully into courtship rhythms, which have been shown to depend on DN1s (see Fujii and Amrein, 2010).

We have made the textual changes to accurately represent the DN1 experiments in Fujii and Amrein, 2010 (e.g., line 255 and 285-286).

Minor points:

6. Figure 1. The authors report that sleep deprivation (SD) for 8 hours or 12 hours between ZT12-ZT0 had no effect on sleep, while 12 hours SD between ZT16-ZT4 or 16 hours SD had an effect on male courtship. They also report that 16 hrs SD, but not 8 hrs SD produced sleep rebound. Do all conditions that induce courtship suppression also induce sleep rebound? This may be important to understand why some sleep deprivation protocols, but not others, are effective in reducing courtship drive.

We thank the reviewers for this observation. 8-hr SD produces a very small rebound indistinguishable from non-sleep deprived controls and this protocol does not induce courtship suppression. Hence, sleep-deprivation that is strong and produces a significant rebound is responsible for courtship suppression. We have made these observations and experimental results more explicit.

7. Figure 1. The null hypothesis here has to be that the two behaviors do not directly interact (courtship and sleep). Even if this is the case, flies will still not perform courtship behavior while sleeping (i.e. the two are mutually exclusive). The courtship assay immediately follows sleep deprivation, but from the data it would be hard to establish if during this assay the flies are awake. It would be important to show that, while flies court much less, they do indeed move around the chamber.

We now analyzed locomotor activity of individually sleep-deprived males as well as control males, and we found that sleep deprivation did decrease average velocity of males, but these males do move around. These additional results are now added as supplemental figure 2.

8. Page 7 line 23. The statement that "The results suggest that sexual arousal positively regulates female sleep" is contradictory. The effect on arousal of pC1 and pCd can only be an inference; a more direct statement of the observed result should be used instead.

We have made changes in the text describing these results and toned down the statement describing pC1 and pCd effects on sleep (line: 127-133).

9. Page 10 line 11. "The above results identify a neural substrate for sexual arousal to

regulate sleep.” I think is more appropriate to say that forced activation of P1 can overcome sleep drive by activating wake-promoting DN1s. At this point no evidence has been shown that sleep drive has an effect on P1s or DN1s (the authors report that acute silencing of DN1s has no effect on sleep and do not show acute inactivation of P1s see above)

We understand that the dTRPA1 activation can be defined as induced activation of P1 neurons but we have strong experimental support for the argument that “The above results identify a neural substrate for sexual arousal to regulate sleep”. We show that P1 neurons have diminished activity (spontaneous neuronal activity recordings using Arclight, Figure 7) in sleep-deprived animals as compared to the sleep-replete controls. Further, P1 neuron inhibition by TNT and Shibire ts1 increases sleep amount as compared to controls.

DN1 neurons have been previously shown to have altered neural activity at dawn when flies awaken and that DN1 neurons have a wake-regulating effect mediated by the neuropeptide Dh31 (Kunst 2015).

10. Page 12 line 16. “We also found that females pan-neuronally expressing microRNAs targeting fruM (c155>UAS-fruMi) sleep as much as males of the same genotype, while control females expressing a scrambled version (UAS-fruMiScr) sleep less than males (Fig. 6b and 6c).” Has this tool been published before? If so the publication should be cited, otherwise essential controls on the efficacy of this reagent should be included.

We thank the reviewer for this notice, and yes, this tool has been validated and published in Meissner et al, PNAS, and we added this reference now in the main text (we previously added this reference in the method section).

Reviewer #3 (Remarks to the Author):

The following study addresses interplay between sleep and mating behavior at the level of single genes and circuits in fruit flies. Study offers interesting findings that will be of interest for the wide audience. However before recommending this work for publication several questions/comments need to be addressed by authors.

We would like to thank the reviewer for acknowledging the significance of the study as a topic of interest to wide audience.

1. Authors like to frame their findings in terms of action selection for sleep and courtship behavior yet sleep is not classically defined as action but rather as state.

We thank the reviewer for this comment and we agree that the term action selection has been used for different contexts in the existing literature and can be potentially misinterpreted. We are now referring to this choice between sleep and courtship as

behavioral choice and have made the textual edits.

2. In Fig 1a authors make a point that not only duration of the sleep but also timing of the sleep is important for vigor of the male courtship. Yet in Fig 1a there is no statistical comparison between ZT0 and ZT4 for the same 12 hr sleep deprivation.

In our manuscript, we actually compared 12-hour sleep-deprived males with control sleep-replete males, and found that there is no significant difference when SD started at ZT0, but there is a significant reduction of courtship when SD started at ZT4, and these statistical data is now clearly added into the figure.

3. In Fig 1b-e it is not clear how authors take the baseline. How do they exactly choose the time period for baseline?

The start point of baseline was chosen at the same ZT the day before SD. We have added this to the figure legend.

4. For supplementary Fig 1e, authors show that female flies do not have sleep rebound. Is it a known fact? If not authors should state that it is their discovery, otherwise provide reference.

We think the reviewer may have misunderstood our data, as supplementary Fig. 1c and Fig. 1e do indicate that female flies have sleep rebound.

5. In Fig 2a. It would be nice to show high-resolution cellular imaging of these neurons.

The high-resolution pictures of these neurons are published already, e.g., in Zhou et al., 2014. We have used the same genetic strategy to target these neurons and the confocal image showing these neurons in 2a are maximum intensity projects. We are happy to provide the raw confocal data if needed.

6. On page 7, lines 7-9 authors state that “Furthermore, sleep reduction in P1-activated male flies is not due to changes in general locomotor activity, as the experimental and control genotypes are equally active while awake (Supplementary Fig. 3).” Yet locomotor activity is measured with the same tools that measures sleep (beam crossing), however it would be better to evaluate movement changes in these flies using alternative method like simple video tracking and getting the speed and distance data from individual flies. This way authors can monitor small changes in movement of animals that may have changed in experimental group.

We understand the reviewer’s concern and agree that the single beam IR method used for sleep analysis is not perfect to measure locomotor behavior. We have performed detailed video analysis for over 24 hours of P1 activation and find that P1 activation at 27°C does not induce courtship, but increases average velocity by ~50%, and inhibits sleep significantly. Activation of P1 neurons using higher temperatures (28.5°C and 30°C) induces courtship, and increased velocity by over 5 times as expected of courting males. These new results have been now reported in Figure 3. We thank the reviewer for this suggestion as it provides novel insight into P1’s role in

arousal and courtship.

7. In Fig 2.b-d control without LexAop2-FlpL is missing.

We intentionally put LexAop2-FlpL into control genotypes to make the control background more similar with the experimental ones. Additionally, the most relevant control for the dTrpA1 experiments is the within genotype control at permissive and restrictive temperature of 21.5 and 28.5 degrees. We have used change in sleep as a result of neuronal activation as a measure to ensure that our studies are controlled properly.

8. Authors assume that activating P1 neurons in males for 36hr period (Fig2b, c) reduces sleep duration. However, such a prolonged activation of neurons might induce plasticity and reduce P1 neuron impact on its synaptic partners. Therefore, if authors want to claim that activation of P1 neuron increases sexual drive they need to also check male courtship behavior at the beginning and at the end of 36hr period.

We think this is a valid concern in studies requiring prolonged activation of neurons. To address this concern, we analyzed courtship and velocity of males with P1 neurons activated for 24 hours using video tracking. We find that throughout this duration the flies exhibit increase in velocity and wing extension characteristics of courtship behavior. These results have now been reported in Figure 3. We thank the reviewer for this experimental suggestion.

9. Based on Fig4b,c authors claim that sleep deprivation reduces activity of P1 neurons. However, they plot standard error and power spectrum of the imaged signal. It was not clear to me why authors choose to use these measures. More direct way of measuring the change in these flies would be to plot mean of $\Delta F/F$ with standard error in control and experimental flies.

We have used ArcLight, an optical sensor of action potential that has been developed and well characterized by the Nitabach lab, to record P1 activity in intact alive male flies, instead of using the Calcium sensor GCamp. ArcLight is measuring spontaneous neural activity and is capturing faster signals (rise and fall) so calculating the power spectrum is the more accurate way of plotting these data. This rise and fall indicative of voltage changes is occurring at different frequencies hence averaging these data to plot $\Delta F/F$ will represent these data inaccurately. We have added plots showing $\Delta F/F$ vs time comparing sleep-deprived and non-sleep-deprived flies for each preparation. Additionally we have averaged the maximum $\Delta F/F$ for these groups and performed non parametric analysis (Mann-Whitney U test) (figure S9).

10. Based on results of knockdown of dVAcHT gene expression authors conclude that this neurotransmitter is mediating P1 activity on its synaptic partners, yet they never showed that the gene knockdown indeed took place. Authors should use immunohistochemistry or RNA labeling tools to make sure that the gene was really expressed at lower levels than in controls.

This is a common concern for RNAi lines and we agree that the efficacy of the genetic reagent is critical to the experimental result. We have now performed qPCR

studies measuring the mRNA/transcript levels of RNAi lines targeting Ace and VAcHT (Supplementary Figure 4). Based on these results its evident that these two RNAi lines are effective in reducing cholinergic transmission.

11. In Fig 5a most of the gene knockdown may reach statistical significance compared to the controls. Could authors maybe comment on this? Could this be a nonspecific effect of the RNAi knockdown?

Activation of P1 neurons leads to a strong decline in sleep amounts. When this manipulation is paired with RNAi knockdown of different genes involved in neurotransmitter synthesis or release we find a reduction in the P1 neuron induced sleep deficit. This could be a result of adding new mini-white-containing transgenes or off target effects of RNAi manipulation but this effect is still not statistically significant.

On the contrary we find that knockdown of genes involved in vesicle release and ACh neurotransmission strongly inhibit P1 induced sleep loss. Here we report the conservative, maximum adjusted p-value to avoid false positives.

Co-staining experiments between fruLexA and Cha-Gal4 in males and females (no P1 neurons) further support the result from RNAi knockdown experiments. We have also presented co-staining experiments between fruLexA and other neurotransmitter Gal4 lines (Tdc2-Octopamine, Th-dopamine, Trh-serotonin, vGAT-glutamate and GABA) (Figure S5).

12. Authors nicely demonstrate that fruM is necessary for sleep regulation in DN1 neurons. It would be desirable to see the sufficiency experiments too. Authors could express fruM in female DN1 neurons and see if they can convert female sleep into male.

In Fig.8g we showed that fru Δ tra females that express FruM sleep similarly to control females, which clearly indicate that FruM is not sufficient to convert female sleep to male. We suspect the role of other sex specific genes like DsxF (female specific). Indeed, we found that DsxF regulates female sleep as shown in Fig. 8e.

Reviewers' Comments:

Reviewer #1:

Remarks to the Author:

Overall, the authors have improved the manuscript with two new experiments. They find that when P1 neurons are activated (via TrpA1) at lower levels (27C), males show deficits in sleep without a big increase in velocity and without wing extension – whereas at higher temps, they get all 3 behaviors (sleep deficits, increased velocity, and wing extension), suggesting that these behaviors can be separated. This is nice. The authors also now show that there is reciprocal functional connectivity between P1 and DN1, which helps with understanding the mechanism. However, many of the reviewer comments previously related to the author's interpretation of their experiments or writing and organization of the manuscript. The authors have done a minimal job at improving the manuscript along these lines – for each concern, they have added a couple of words or lines of text, but have not undertaken the full scale rewrite that was needed. As an example, one concern raised was about interpreting experiments in which P1 neurons are activated, which causes a reduction in sleep. The authors conclude (still) that increased sexual arousal leads to a lack of sleep (“these results clearly indicate that increased sexual arousal induced by P1 activation suppresses sleep in male flies...”). They never show directly that increased sexual arousal suppresses sleep – nonetheless, they have kept strong statements like this (and others) in the manuscript. They now provide new data that lower levels of P1 activation suppress sleep without affecting wing extension (and presumably sexual arousal?)...so by their own (new) data this is probably not the correct interpretation. Another example concerns their insistence to keep the intro focused on action selection (they change the terminology now to “behavioral choice” which is still not correct) despite reviewer criticisms that what they are really studying is the interaction between potentially competing drives (sleep and courtship), but since they don't know the precise relationship between P1 and courtship drive, they should avoid making such strong statements in their Intro. These kinds of misstatements and overstatements that persist in the manuscript diminish my enthusiasm somewhat, in spite of the new experimental data.

Reviewer #2:

Remarks to the Author:

The authors appropriately engaged with previous criticism and suggestions, and made a strong effort to address all of this reviewer's comments. The paper now reads much better and is more cohesive. I have no further major criticism.

minor corrections/clarifications:

196 - this statement could be clarified

220-221 - I am surprised that flies with ubiquitous knock-down of Ace and VAcH1 are viable-
are these non-essential genes?

305-306 "We also analyzed these results using another way and found the same conclusions".
This sentence should probably be re-written

Reviewer #3:

Remarks to the Author:

After reading authors responses and updated version of the manuscript I do not have further questions and would like to recommend it for publication.

Our responses to reviewer comments are highlighted in red.

Reviewer #1 (Remarks to the Author):

Overall, the authors have improved the manuscript with two new experiments. They find that when P1 neurons are activated (via TrpA1) at lower levels (27C), males show deficits in sleep without a big increase in velocity and without wing extension – whereas at higher temps, they get all 3 behaviors (sleep deficits, increased velocity, and wing extension), suggesting that these behaviors can be separated. This is nice. The authors also now show that there is reciprocal functional connectivity between P1 and DN1, which helps with understanding the mechanism. However, many of the reviewer comments previously related to the author's interpretation of their experiments or writing and organization of the manuscript. The authors have done a minimal job at improving the manuscript along these lines – for each concern, they have added a couple of words or lines of text, but have not undertaken the full-scale rewrite that was needed. As an example, one concern raised was about interpreting experiments in which P1 neurons are activated, which causes a reduction in sleep. The authors conclude (still) that increased sexual arousal leads to a lack of sleep (“these results clearly indicate that increased sexual arousal induced by P1 activation suppresses sleep in male flies...”). They never show directly that increased sexual arousal suppresses sleep – nonetheless, they have kept strong statements like this (and others) in the manuscript. They now provide new data that lower levels of P1 activation suppress sleep without affecting wing extension (and presumably sexual arousal?)...so by their own (new) data this is probably not the correct interpretation. Another example concerns their insistence to keep the intro focused on action selection (they change the terminology now to “behavioral choice” which is still not correct) despite reviewer criticisms that what they are really studying is the interaction between potentially competing drives (sleep and courtship), but since they don't know the precise relationship between P1 and courtship drive, they should avoid making such strong statements in their Intro. These kinds of misstatements and overstatements that persist in the manuscript diminish my enthusiasm somewhat, in spite of the new experimental data.

We are gratified that the reviewer acknowledges that the new experiments showing the mutually excitatory interactions between P1 and DN1 neurons provides a mechanistic understanding of how sleep and sex are co-regulated.

We appreciate the reviewer's suggestion on the writing of P1's role in sleep regulation, and have made textual changes (in track change mode). Our new data show that P1 neurons regulate sleep and courtship in a threshold-dependent manner, and we focused our writing on it.

We also made substantial textual changes in the abstract, introduction, and other parts of the manuscript where necessary, to avoid the use of terms “behavioral prioritization/action selection” as suggested. In occasional places such as the beginning of the introduction we still used “behavioral choice/decision-making” to frame our work and bring out the central question studied in this manuscript, and these terms are also minimally used. We modified the introduction to clearly state our motivation for conducting the study to understand interactions between competing behaviors (sleep and sexual behaviors) in genetic and neuronal levels. We also change the title to “Genetic and neuronal mechanisms governing the sex-specific interaction between sleep and sexual behaviors in *Drosophila*”.

Reviewer #2 (Remarks to the Author):

The authors appropriately engaged with previous criticism and suggestions, and made a strong effort to address all of this reviewer's comments. The paper now reads much better and is more cohesive. I have no further major criticism.

We would like to thank the reviewer for noticing the significant rewrite and experimental effort made by us.

minor corrections/clarifications:

196 - this statement could be clarified

We now made a more specific statement (now p282: which may be due to different populations and/or numbers of P1 neurons targeted by these two methods).

220-221 - I am surprised that flies with ubiquitous knock-down of Ace and VAcHT are viable- are these non-essential genes?

Flies with Ace knock-down are fully viable, while flies with VAcHT knock-down are partially lethal. The survival could be a result of incomplete knock-down as indicated by our qPCR results. We now added a brief description of this into the supplementary figure legend.

305-306 "We also analyzed these results using another way and found the same conclusions". This sentence should probably be re-written

We have modified this statement to accurately reflect the comparison between $\Delta F/F_0$ of P1 neuron activity in sleep-deprived and sleep replete flies (now p401: We also analyzed these results using comparisons between peak or maximal $\Delta F/F_0$ and found significant differences).

Reviewer #3 (Remarks to the Author):

After reading authors responses and updated version of the manuscript I do not have further questions and would like to recommend it for publication.